# Reactive chlorine, sulphur and nitrogen containing volatile organic compounds impact atmospheric chemistry in the megacity of Delhi during both clean and extremely polluted seasons

Sachin Mishra[1], Vinayak Sinha[1], Haseeb Hakkim[1], Arpit Awasthi[1], Sachin D. Ghude[2], Vijay Kumar Soni[3], Narendra Nigam[3], Baerbel Sinha[1], Madhavan N. Rajeevan[4]

[1] Department of Earth and Environmental Sciences, Indian Institute of Science Education and Research Mohali, Sector 81, S.A.S Nagar, Manauli PO, Punjab, 140306, India

[2]Indian Institute of Tropical Meteorology, Pashan, Pune 411 008, Ministry of Earth Sciences, India

[3]India Meteorological Department, New Delhi 110 003, India, Ministry of Earth Sciences, India

[4]Ministry of Earth Sciences, Government of India, New Delhi 110 003, India

[*] *Correspondence to*: Vinayak Sinha (vsinha@iisermohali.ac.in)

**Abstract.** Volatile organic compounds significantly impact the atmospheric chemistry of polluted megacities. Delhi is a dynamically changing megacity and yet our knowledge of its ambient VOC composition and chemistry is limited to few studies conducted mainly in winter before 2020 (all pre-COVID). Here, using a new extended volatility range high mass resolution (10000-15000) Proton Transfer Reaction Time of Flight Mass Spectrometer, we measured and analyzed ambient VOC-mass spectra acquired continuously over a four-month period covering "clean" monsoon (July-September) and "polluted" post-monsoon seasons, for the year 2022. Out of 1126 peaks, 111 VOC species were identified unambiguously. Averaged total mass concentrations reached $\sim260\mu gm^{-3}$ and were >4 times in polluted season relative to cleaner season, driven by enhanced emissions from biomass burning and reduced atmospheric ventilation (~2). Among 111, 56 were oxygenated, 10 contained nitrogen, 2 chlorine, 1 sulphur and 42 were pure hydrocarbons. VOC levels during polluted periods were significantly higher than most developed world megacities. Methanethiol, dichlorobenzenes, C6-amides and C9-organic acids/esters, which have previously never been reported in India, were detected in both the clean and polluted periods. The sources were industrial for methanethiol and dichlorobenzenes, purely photochemical for the C6-amides and multiphase oxidation and partitioning for C9-organic acids. Aromatic VOC/CO emission ratio analyses indicated additional biomass combustion/industrial sources in post-monsoon season, alongwith year-round traffic sources in both seasons. Overall, the unprecedented new information concerning ambient VOC speciation, abundance, variability and emission characteristics during contrasting seasons significantly advances current atmospheric composition understanding of highly polluted urban atmospheric environments like Delhi.

## 1 Introduction

The national capital territory of Delhi in India is jointly administered by the central and state governments and accommodated more than 32 million people in 2022. For the past several years, its population has grown at the rate of more than 2.7 percent per year, adding about 1 million new inhabitants each year. Thus, the region

represents a complex dynamically changing emission environment driven by rapid changes in emissions as regulatory authorities make efforts to improve urban infrastructure and public transportation while promoting cleaner technologies. As a megacity in a developing country with one of the world's highest population densities, Delhi exemplifies some of the key challenges faced by many megacities in the global south, where increased urbanization and inequitable access to clean energy sources along with unfavourable meteorological conditions during cold periods of the year, cause the inhabitants to suffer from extreme air pollution episodes. Lelieveld et al. (2015) identified South Asia as one of the global air pollution hotspots in terms of the contribution of outdoor air pollution sources to premature mortality due to particulate matter pollution. Reduction of other atmospheric pollutants is also deemed necessary to fulfil the UN Sustainable Development Goals (Keywood et al., 2023). Thus, the study of Delhi's ambient chemical composition using state of the art technology can offer valuable insights and lessons for our understanding of polluted atmospheric environments.

Previous studies have demonstrated that air pollution in the Delhi-NCR metropolitan area peaks during the post-monsoon (October- November) season (e.g. Kulkarni et al., 2020), coinciding with the time of year when large scale paddy stubble burning occurs in the Indo-Gangetic Plain (Kumar et al., 2021). The main air pollutant in exceedance has long been identified to be particulate matter (e.g. $PM_{2.5}$) and many studies (Gani et al., 2020; Cash et al., 2021; Sharma et al., 2023; Singh et al., 2011) have documented the variability, exceedance and composition of aerosols. Volatile organic compounds (VOCs) are major precursors of secondary organic aerosol, which is a significant component of $PM_{2.5}$ (30-60% in Delhi; Chen et al., 2022; Nault et al., 2021) and surface ozone over Delhi. In fact, in-situ ozone production in Delhi has been reported to be more sensitive to VOCs rather than nitrogen oxides (Nelson et al., 2021). Several VOCs (e.g. benzene, nitromethane, 1,3-butadiene) are also carcinogenic (WHO 2010) at high exposure concentrations and many pose direct health risks (Ho et al., 2006; Espenship et al., 2019; WHO 2019; Weng et al., 2009; Roberts et al., 2011; Durmusoglu et al., 2010). VOCs can also aid source apportionment studies by acting as source fingerprints and valuable molecular markers of specific emission sources (de Gouw et al., 2017; Holzinger et al., 1999; Warneke et al., 2001; Kumar et. al., 2020; Garg et al., 2016; Hakkim et al; 2021; Kumar et al., 2021). In the complex emission environment of cities in the developing world, this can be especially helpful since the energy usage portfolio is such that biomass burning sources are likely to be as significant as fossil-fuel based sources (Bikkina et al., 2019) in influencing the air pollutant burden of VOCs, resulting in ambient air VOC composition that could be quite different from cities like Los Angeles (McDonald et al., 2018).

Existing knowledge about the abundance and diurnal variability of major ambient VOCs such as methanol, acetone, acetaldehyde, acetonitrile, isoprene, benzene, toluene, xylenes and trimethyl benzenes in Delhi, is limited to just four previously measured wintertime datasets: Dec-March of 2016 (Chandra et al., 2018; Hakkim et al., 2019), Dec-March of 2018 (Wang et al., 2020; Tripathi et al., 2022), few days in October 2018 (Nelson et al., 2021; Bryant et al., 2023) and one spanning 145 days of 2019 that reported source apportionment of some VOCs for different seasons (Jain et al., 2022). We note that all these were pre-COVID period datasets, and that since these observations many new regulations have been put in place e.g. for traffic with the introduction of BS-VI (EURO6 equivalent) in 2020 and the Faster Adoption and Manufacturing of hybrid and Electric vehicles (FAME) program for promotion of E-vehicles, and for industries with a ban on the use of petcoke in the National Capital Region (NCR) and the crackdown on unregistered industries (Guttikunda et al, 2023). After COVID lock-downs happened in 2020, a new Commission for Air Quality Management in Delhi National Capital Region and its

Adjoining Areas (CAQM) was set up in November 2020 (https://caqm.nic.in/index.aspx?langid=1 ). Under its mandate, depending on air quality level, it promulgates immediate graded response action plans (GRAP; https://caqm.nic.in/index1.aspx?lsid=4168&lev=2&lid=4171&langid=1) that instruct civic authorities to shut-down or restrict particular emission sources. Furthermore, on 7 August 2020, the Delhi government announced a new Delhi Electric Vehicle (EV) Policy. In order to address the high-upfront cost of EVs (ICE vehicles), the Delhi EV Policy provides demand incentives for purchasing electric vehicles. The incentives help bring cost parity for EVs and are in addition to those outlined in the Faster Adoption and Manufacturing of Hybrid and Electric Vehicles (FAME II). In the budget allocation for 2020, the Government of India allocated $600 million USD for clean air measures through the Ministry of Housing and Urban Affairs (MoHUA) to 46 cities across India. These have been detailed in a report by Arpan Chatterji (2020). Thus overall, important changes to the transport emission sector, construction and urban industrial sector and residential sector were implemented at a policy level after 2020 to reduce air pollution in the Delhi-NCR region.

The monsoon season which precedes the post-monsoon season lasts from June to September and is characterized by better air quality, aided by favourable meteorological conditions, including higher ventilation co-efficient, negligible agricultural waste burning and enhanced wet scavenging (Kumar et al., 2016).

This study addresses some of the above knowledge gaps pertaining to ambient VOCs during the "clean" monsoon season characterized by baseline pollution levels and the polluted "post-monsoon" season characterized by extreme pollution events and large scale open agricultural biomass waste fires regionally. Employing a new extended volatility range (EVR) high mass resolution (10000-15000) Proton Transfer Reaction Time of Flight Mass Spectrometer 10K (PTR-TOF 10000; Ionicon Analytik GmBH), a technology that has never before been deployed in India, we investigated the ambient VOC speciation, abundance, variability and emission characteristics in the polluted urban environment of Delhi over a 4-month period. This enabled us to discover several low volatility VOCs, many of which are present in fire emissions (Koss et al., 2018), for the time in South Asia, as all previous VOC studies have involved either the older PTR-TOF-MS or PTR-QMS instruments, that have significantly lower mass resolution and lower detection sensitivity and did not possess the extended volatility range components. We first undertook comprehensive and rigorous interpretation of the ambient mass spectra over a four-month period spanning July-Nov of 2022 in Delhi. This was followed by identification and quantification of 111 VOCs, many of which have been discovered and reported for the first time from the South Asian atmospheric environment. Each of these compounds was then classified in terms of oxygenated VOCs, pure hydrocarbons, major nitrogen containing VOCs, chlorine containing VOCs and sulphur containing VOCs, followed by the time series analyses and diurnal profiles of the major VOCs and some new/rarely reported VOCs in both seasons as a function of meteorology and emissions. The atmospheric chemistry implications of some of the newly discovered compounds in this polluted urban environment are discussed. Further, using measured aromatic VOC/CO emission ratios in monsoon and post-monsoon season, a global comparison with reports from megacities in Europe, North America and Asia was undertaken for a nuanced understanding of their levels and sources in Delhi relative to megacities across these different continents.

## 2. Methodology

### 2.1 Measurement site and meteorological conditions:

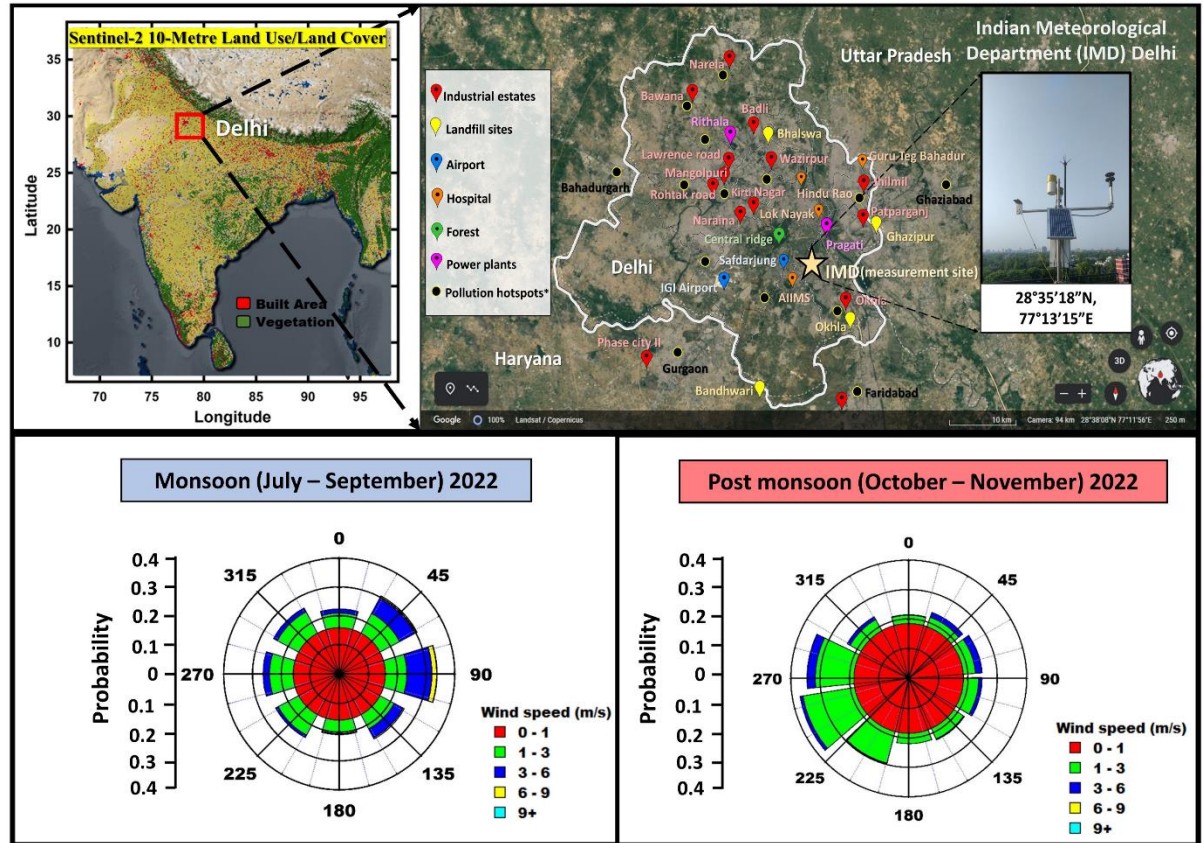

**Figure 1: Map of India showing Delhi (1 a) and zoom in of the measurement site (star marked) (1b; Google Earth Imagery © Google Earth) with a view from the roof-top of the SatMet Building (28.5896°N-77.2210°E), and wind rose plots derived from in-situ one-minute wind speed and wind direction data during monsoon (1c) and post-monsoon (1d) 2022 acquired at sampling height of ~35m A.G.L**

The measurement site was located within the premises of the India Meteorological Department (IMD) which is situated in Central Delhi (Fig. 1). Ambient air was sampled at a height of circa 35m above ground level from the roof-top of the SatMet building (28.5896°N-77.2210°E), into the instruments which were housed inside a laboratory located in the sixth floor of the same building.

Figure 1 (a) shows the land use/ land cover (Sentinel-2 10m) map of India with a red marked box highlighting Delhi. The city is bordered on its northern, western, and southern sides by the state of Haryana and to the east by the state of Uttar Pradesh. The star marked in Fig. 1 (b) shows the measurement site (IMD Delhi) and its surroundings. The major pollution hotspots include places like Ghaziabad (towards the northeast), Bahadurgarh (towards the northwest), Gurgaon (towards the southwest), and Faridabad and Okhla (towards the southeast),. Major industrial areas e.g. Okhla industrial area, major landfill sites ,the international airport and some major hospitals are also shown in Fig 1 (b).

Meteorological sensors (Campbell Scientific Inc.) were deployed to measure the wind speed, direction, temperature, relative humidity and photosynthetic active radiation (model nos.: CS215 for temperature and RH, PAR PQSI sensor, and for rain TE525-L40). Boundary layer height was taken from ERA5 reanalyses dataset (Hersbach et al., 2023) and ventilation coefficient was calculated as the product of the measured wind speed and boundary layer height. Atmospheric ventilation or ventilation coefficient (VC) is a good proxy for the dilution and dispersion of air pollutants near the surface (Hakkim et al., 2019). It is defined as the product of boundary layer height (m) and wind speed ($ms^{-1}$). The VC represents the rate at which air within the mixed layer is transported away from a region of interest and provides information about how concentrations of pollutants are modulated through transport of air over that region. Figures 1 (c) and 1 (d) show the wind rose plot derived from the in-situ one-minute wind speed and wind direction data acquired at the measurement site for monsoon (July 2022 – September 2022) and post-monsoon (October 2022 – November 2022) seasons, respectively. The prevalent wind direction changed from easterly flow in monsoon season to westerly flow in the post-monsoon season. During the monsoon season, the major fetch region spanned from the NE to SE-E. These NE, E, and SE winds were associated with high wind speeds ranging from 3 – 6 $ms^{-1}$, which on occasions reached up to 9 $ms^{-1}$. During the post-monsoon season, the major wind flow was from the NW to the SW-W sector. These wind speeds were lower, ranging from 1 – 3 $ms^{-1}$ exceeding 6 $ms^{-1}$ only occasionally. Overall, the site received air from all wind sectors in both seasons. This is also borne by the back trajectory analyses presented in the companion paper (Awasthi et al., 2024), which showed that the site is characterized by regional airflow patterns as documented at other sites in the Indo-Gangetic Plain (Pawar et al., 2015).

Fire count data were obtained using the Visible Infrared Imaging Radiometer Suite (VIIRS) 375m thermal anomalies/active fire product data from the VIIRS sensor aboard the joint NASA/NOAA Suomi National Polar-orbiting Partnership (Suomi NPP) and NOAA-20 satellites for high and normal confidence intervals only.

## 2.2 Measurement of Volatile Organic Compounds using the PTR-TOF-MS 10K

Volatile organic compounds (VOCs) were measured using a new high sensitivity and high mass resolution Proton Transfer Reaction Time of Flight Mass Spectrometer (PTR-TOF-MS 10k, model PT10-004 manufactured by Ionicon Analytik GmbH, Austria). While PTR-TOF-MS 8000 series (Tripathi et al., 2022) and PTR-QMS (Sinha et al., 2014) instruments have been previously deployed in India and have mass resolutions of 8000 and 1, respectively, this study marks the first deployment of the PTR-TOF-MS 10K system in India, a system that possesses several unique advantages over the older generation instruments for VOC measurements in polluted and complex emission environments. The first is that this new system is equipped with the extended volatility range technology (Piel et al., 2021), ensuring that even many intermediate volatility range compounds and sticky VOCs can be detected with very fast response times and minimal surface effects. The inlet system of the instrument as well as the ionization chamber is fully built into a heated chamber and the inlet capillary is further fed through a heated hose to ensure there are no "cold" spots for condensation. The entire inlet system is made of inert material (e.g. PEEK or siliconert treated steel capillaries to keep surface effects minimal. Additionally, a 7 μm siliconert filter just before the drift tube served to minimize clogging/contamination of the system. The second advantage possessed by the PTR-TOF-10K used in this work is the inclusion of an ion booster funnel and hexapole ion guide placed after the drift tube/reaction chamber for improved extraction of ions in a manner that boosts both the mass resolution as well as the sensitivity over its older peers. This helped achieve much higher mass resolution

(> 10000 m/Δm), even reaching as high as 15000 m/Δm at m/z 330, and detection limits better than 3 ppt for all
compounds detected in the mass to charge ratio (m/z) 31-330 mass range. These customizations over previously
deployed PTR-TOF-MS instruments in Delhi, enabled detection and discovery of several intermediate range-
volatility compounds (IVOCs) in the gas phase. Other parts of the instrument have already been explained well
earlier (Jordan et al., 2009; Graus et al., 2010). During this study, the instrument was operated at a drift tube
pressure of 3 mbar, drift tube temperature of 120 $^0$C, and drift tube voltage of 600V, resulting in an operating E/N
ratio of ∼ 120 Td (1 Td = $10^{-17}$ V cm$^{-2}$). These operational instrumental settings are also summarized in Table S1.
Ambient air was sampled continuously from the rooftop (∼35m A.G.L) through a Teflon inlet line that was
protected with a Teflon membrane particle filter (0.2 μm pore size, 47 mm diameter) to ensure that dust and debris
did not enter the sampling inlet. The length of the inlet line was 5m and made of Teflon (3m 1/8 inch O.D. and
2m 1/4inch O.D). The total inlet residence time was ∼2.7 seconds. The part of the inlet that was indoors (3m of
1/8 inch O.D.) was well insulated and heated to 80 degree Celsius. We think this short inlet residence time and
heated inlet facilitated the detection of IVOCs, relative to previous studies. The instrument background was
acquired regularly (typically every 30 min for 5 min), by sampling VOC-free zero air. VOC-free zero air was
produced by passing air through an activated charcoal scrubber (Supelpure HC, Supelco, Bellemonte, USA) and
a VOC scrubber catalyst (Platinum wool) maintained at 370 $^0$C. Mass spectra covering the m/z 15 to m/z 450
range were obtained at 1 Hz frequency. An internal standard comprising 1,3-di-iodobenzene ($C_6H_5I_2^+$) detected at
m/z 330.848 and its fragment ion [$C_6H_5I^+$]) detected at m/z 204.943 were constantly injected to ensure accurate
mass axis calibration, so that any drifts in the mass scale were corrected providing for accurate peak detection.
Primary data acquisition of mass spectra was accomplished using the ioniTOF software (version 4.2; IONICON
Analytik Ges.m.b.H., 6020 Innsbruck, Austria). All the settings related to PTR (Proton Transfer Reaction), TPS
(TOF power supply), MPV (Multi-port-valve), and MCP (Multi-channel plate) can be controlled and optimized
using this control software. The raw mass spectra and relevant instrumental metadata are stored in HDF5 format.
These spectra were further processed using the Ionicon Data Analytik (IDA version 2.2.0.4; Ionicon Analytik
GmbH, Innsbruck, Austria) software that has the functionalities for peak search, peak fits and preliminary mass
assignments and identification of a broad spectrum of organic compounds. The IDA software employs an
automated peak detection routine guided by user-defined sensitivity levels for peak detection, peak fit, and shape.
The software then uses chemical composition information based on the exact masses and isotopic patterns and
calculates a specific proton transfer rate constant (k-rate) based on the polarizability and dipole moment for the
peaks with an assigned chemical formula, instead of using a generic value as was done in previous PTR-TOF-MS
measurements in Delhi (Tripathi et al., 2022). We manually compared the values also with the compilation of k
rates reported by Pagonis et al., (2019) as an additional check. The user has possibility to define a window for
mass accuracy (e.g. 30 ppm). Within this defined range and accuracy window, the software identifies all possible
chemical compositions and molecular formulae and calculates the corresponding isotope patterns. These patterns
are then compared to find the best-fit chemical composition. The process is carried out iteratively, starting with
the lower m/z values, according to the method described in the study by Stark et al., (2015).
In this study, a total of 1126 peaks were detected in the raw measured ambient mass spectra. After further
additional quality control and assurance steps performed manually as detailed in the Section 3.0, 111 compounds
present in ambient air for which the molecular formula could be confirmed unambiguously are reported and for
which isotopologues due to molecules of different chemical composition could be ruled out completely, were

further analysed in this work. The term "unambiguous" is used in the context of the accurate elemental composition/molecular formula assignment of the ions by leveraging the high mass resolution (8000-13000 over entire dynamic mass range) and detection sensitivity (reaching even 1 ppt or better for many ions; see Table S2) of the instrument. This enabled ensuring peaks due to expected isotopic signals were not construed as new compounds if their height was exactly as expected for a shoulder isotopic peak based on the natural abundance of isotopes of carbon, hydrogen, nitrogen, sulphur, chlorine and oxygen that made up the more abundant molecular ion. Where an ion could occur significantly due to fragmentation of another compound, the same has also been noted in Table S2 during attribution of the compound's name. Figure S1 provides an example of visualization of mass spectra and peak assignment using the IDA software which also illustrate the high mass resolving power of the PTR-ToF-MS 10K, that enables separation of ion signals that differ by less than 0.04 Th, as well the identification of isotopic peaks of parent compounds like methanethiol, dichlorobenzene, C-6 amide and C-9 carboxylic acid acid (Fig S2), which are discussed in detail in Section 2.4. Table S2 also provides the limit of detection (LoD) of the compounds as well as the average and interquartile range observed season-wise for each ion. The LoD was calculated by taking the $2\sigma$ value of the VOC-free zero air instrument background (Müller et al., 2014). Example of measured data showing the instrumental backgrounds and ambient levels for methanethiol, dichlorobenzene, C-6 amide and C-9 carboxylic acid , over a 3h period are illustrated in Fig S3. A certified VOC calibration gas mixture (Societa Italiana Acetilene E Derviati; S.I.A.D. S.p.A., Italy) containing 11 hydrocarbons at ~100 ppb, namely methanol, acetonitrile, acetone, isoprene, benzene, toluene, xylene, trimethylbenzene, and dichlorobenzene and trichlorobenzene was used during the field deployment for measuring the transmission and sensitivity of compounds covering the mass range (m/z=33 to m/z = 181). The instrument was calibrated a total of 8 times during the study period: 21.07.2022 after first installation, 26.09.2022, 21.10.2022, 26.10.2022, 5.11.2022, 11.11.2022, 16.11.2022 and 30.11.2022. Results were reproducible (~21% or better for all compounds) across all experiments and a transmission curve obtained from one of the calibration experiments is shown in Fig. S4. Measured transmission further allowed for more accurate quantification by accounting for correction of the mass-dependent detection efficiency of the system. Equation S1 (de Gouw et al., 2007) was then used to convert the measured ion signals to mixing ratios. The linearity for compounds available in the VOC standard were also checked independently and was above $r \geq 0.9$ as illustrated in Fig S5 for the tested range of ~2 to 8 ppb. The background corrected concentrations of all the detected m/z were exported from IDA in .csv format and further analysis of the dataset was carried out using IGOR Pro software (version 6.37; WaveMetrics, Inc.). The overall uncertainty calculated using the root mean square propagation of errors due to the accuracy of gas standard and flow controllers was ~13 % or better for compounds present in the VOC gas standard. For other compounds reported in this work, it is estimated that the combined accuracy of the transmission function and the parameterized k-rates, put the overall uncertainty in the range of ±30% (Reinecke et al., 2024).

Carbon monoxide (CO) was measured using IR filter correlation-based spectroscopy air quality analyzer (Thermo Fischer Scientific 48i) while ozone was measured using UV absorption photometry (Model 49i; Thermo Fischer Scientific, Franklin, USA). The overall uncertainty of the measurements was less than 6%. Details concerning characterization of the instrument including calibration and data QA/QC protocols have been comprehensively described in our previous works (Chandra and Sinha, 2016; Kumar et al., 2016; Sinha et al., 2014).

**2.3 Mass assignment and compound identification**

A total of 1126 peaks were detected in the raw mass spectra. To identify the ambient compounds of relevance in Delhi from these detected peaks, the following additional manual quality control checks were undertaken. First, peaks attributed to non-ambient compounds such as the impurity ions (e.g. $NO^+$), water cluster ion peaks, and peaks associated with internal standards were excluded resulting in 1025 peaks for further consideration. Next, the diel profiles and detection limits of these 1025 ion peaks were perused. Only 319 ions out of the 1025 ions showed some diurnal variability and had values above the detection limit after accounting for the respective instrumental background. Next, we verified the presence and expected theoretical magnitude of the shoulder isotopic peaks based on the natural isotopic distribution abundance of the elemental composition of the ion. Fig S6 provides a visual example. This was feasible for all m/z except the C1 oxygen containing analyte ions, where the shoulder peak was below detection limit. The preceding QA/QC resulted in an unambiguous assignment for 111 of the 319 ions. Note that these 111 explained 86% of the total mass concentration ($\mu gm^{-3}$) observed due to the 319 detected peaks when accounting for the isotopic peaks as well. Table S2 lists the ion m/z and molecular formula of the corresponding compound, along with the averaged mixing ratios observed in each case during the monsoon and post-monsoon season. Additionally, the characteristic ambient diel profile classification as one of the following: unimodal with daytime peak for biogenic/ evaporative/ photochemical source emitted compounds, bimodal with morning and evening peaks for compounds driven by primary emissions (e.g. toluene) and trimodal which were hybrid of the former two, are also provided for each species. Compound names were attributed to specific ions using assignments reported at that m/z in the compiled peer-reviewed PTR-MS mass libraries published by Yáñez-Serrano et al., (2021) and Pagonis et. al., (2019) as well as previously published pioneering reports by Stockwell et. al. (2015), Sarkar et al. (2016), Yuan et al. (2017) and Hatch et al. (2017).

Fragmentation of certain compounds in specific atmospheric environments can cause significant interferences in the detection of major compounds like isoprene, acetaldehyde and benzene, as reported recently by Coggon et al., 2024. We checked these as well as an additional quality control measure. As noted by Coggon et al. 2024, isoprene can suffer significant interferences from higher aldehydes as well as substituted cyclohexanes, which can fragment and add to the signal at m/z 69.067 (at which protonated isoprene $C_5H_9^+$ is also detected). The magnitude depends on the instrument operating conditions (Townsend ratio), instrument design and the mixture of VOCs present in ambient air while co-sampling isoprene. Coggon et al. 2024 very nicely clarified both these aspects and found that when influenced by cooking emissions and oil and natural gas emissions and at higher Townsend ratios, these interferences can be quite significant and even account for upto 50% of the measured signal attributed to isoprene in extreme cases. We operated the PTR-TOF-MS at 120 Td which minimizes fragmentation even if it occurs, compared to when operated at 135-140 Td. Concerning the ambient VOC mixture and emission sources, we note that the type of restaurant cooking emissions present in Las Vegas and over Oil and Natural Gas petrochemical facilities in USA for which Coggon et al. 2024 reported the highest isoprene interferences, were absent/negligible at the study site in Delhi. In the latter, open biomass burning sources such as paddy residue burning in post-monsoon season and garbage biomass fires and traffic that occur throughout the year are more significant. Use of more specific though slower analytical techniques based on gas chromatography show that such biomass combustion sources emit significant amounts of isoprene (Andrea et al., 2019; Kumar et al., 2021). The above points and supporting TD-GC-FID measurements of isoprene, benzene and toluene (see Fig S7 and Shabin et al., 2024), led us to conclude that such correction is unwarranted for our PTR-TOF-MS dataset. Concerning the

interference on acetaldehyde detection due to ethanol, we note that even in Coggon et al. 2024 this was reported to only be of significance in highly concentrated ethanol plumes such as those encountered on the Las-Vegas strip where ~1500 ppb of ethanol was detected. On the contrary, in Delhi as listed in Table S2, ethanol values detected at m/z 47.076 were on average only 0.2 ppb (Interquartile range 0.16 ppb) during monsoon and 0.55 ppb (Interquartile range 0.5 ppb) in post-monsoon season, respectively, whereas acetaldehyde detected at m/z 45.03 was significantly higher at 3.34 and 7.75 ppb during monsoon and post-monsoon season, respectively.

For the same molecular formula, several isomeric compounds with differing chemical structures are possible, with the number of possibilities increasing enormously with an increase in the number of atoms that make up the molecule. In addition, in some instances fragmentation of other compounds can complicate the compound attribution for a given ion. Nonetheless in the interest of stimulating interest and further investigation as many have been previously rarely reported or are being reported for the first time in ambient air, we have made bold to provide one of the many possible chemical structures in the Table S2. We do caution that the chemical structure provided by no means even constitutes a best guess estimate but nonetheless would be appealing to chemists and provoke further detailed reporting rather than just the molecular formula.

## 3. Result and Discussion:

### 3.1: Analyses of ambient mass spectra and mass concentration contributions of VOC chemical classes

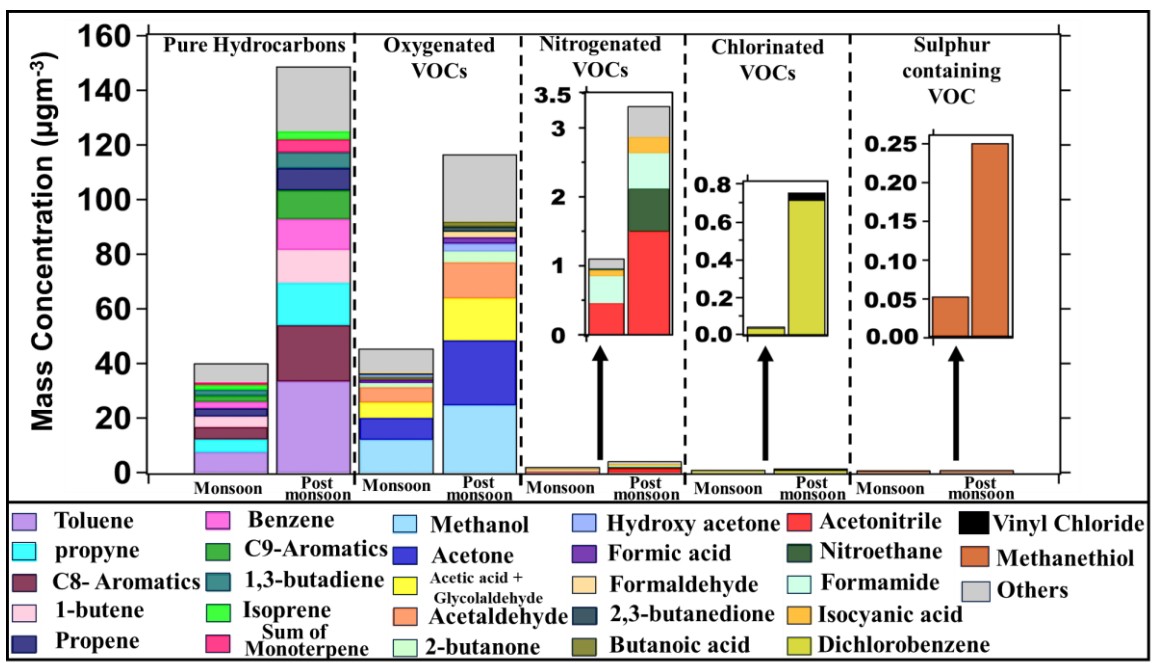

**Figure 2: Bar graph of 111 compounds class-wise, namely Pure Hydrocarbons, Oxygenated VOCs (OVOCs), Nitrogen-containing VOCs (NVOCs), Chlorine-containing VOCs (ClVOCs), and sulphur-containing VOC (SVOC) in both monsoon and post-monsoon periods.**

A summary of the distribution of the 111 compounds in terms of chemical classes showing their averaged measured ambient mass concentration ($\mu gm^{-3}$) contributions is shown in Fig. 2 for the monsoon (22$^{nd}$ July – 30$^{th}$

September 2022) and post-monsoon seasons (1 October- 26 November 2022). Out of the 111 compounds, 42 were pure hydrocarbons made up only of carbon and hydrogen atoms, 56 were oxygenated volatile organic compounds (OVOCs) made up of only carbon, hydrogen and oxygen, 10 contained nitrogen (NVOCs), 2 contained chlorine (ClVOCs), and 1 contained sulphur (SVOC). The average total mass concentration of the same set of pure hydrocarbons during post-monsoon season was 3.7 times greater than in monsoon season (40 $\mu$gm$^{-3}$ vs 148 $\mu$gm$^{-3}$) while the average total mass concentration of OVOCs during post-monsoon was 2.6 times greater than the monsoon season values (44 $\mu$gm$^{-3}$ vs 116 $\mu$gm$^{-3}$). Pure hydrocarbons and OVOCs contributed similarly to the mass concentrations in monsoon season but during the post-monsoon season, the contribution of pure hydrocarbons was significantly higher than that of OVOCs, due to an increase in primary emissions of these compounds. The average mass concentration of NVOCs during post-monsoon was thrice as high relative to the monsoon season (1 $\mu$gm$^{-3}$ and 3 $\mu$gm$^{-3}$). For the chlorine containing VOCs the post-monsoon, concentrations were 20 times higher, though in absolute magnitude, the values were low (1 $\mu$gm$^{-3}$). The average mass concentration of sulphur containing VOCs during post-monsoon was 4 times higher, but again absolute values were low (0.2 $\mu$gm$^{-3}$). The top 10 pure hydrocarbon compounds by mass concentration ranking were toluene, sum of C8-aromatics (xylene and ethylbenzene isomers), propyne, 1-butene, benzene, sum of C9-aromatics (trimethyl benzene isomers), propene, sum of monoterpenes, isoprene and 1,3 butadiene and contributed to 84% of the total mass concentration due to pure hydrocarbons during both the monsoon and post-monsoon seasons, respectively, while the top 20 contributed to 95% and 96% of the total mass concentration in monsoon and post-monsoon, respectively. The top 10 OVOCs: methanol, acetone, acetic acid+ glycolaldehyde, acetaldehyde, hydroxyl-acetone, formaldehyde, 2-butanone, 2,3-butanedione, formic acid, butanoic acid collectively contributed to 84% and 79% of the total mass concentration due to all OVOCs in monsoon and post-monsoon, respectively, while the top 20 contributed to 93% and 90% of the total mass concentration in monsoon and post-monsoon, respectively. The top 4 NVOCs namely acetonitrile, nitroethane, formamide and isocyanic acid contributed to 92% and 91% of the total mass concentration in monsoon and post-monsoon, respectively. Out of 2 identified chlorine containing VOCs, dichlorobenzene ($C_6H_4Cl_2$) was found to be the major contributor contributing 87% and 95% of the total mass concentration in monsoon and post-monsoon, respectively. The only sulphur containing VOC was methanethiol [$CH_4S$] detected at its protonated ion m/z 49.007 and confirmed by the shoulder isotopic peak. Overall, there was an increase in the mass concentration of all the classes of VOCs from monsoon to post-monsoon. This increase in mass concentration could be attributed to increased emissions from sources that get active in post-monsoon, such as regional post-harvest paddy residue burning, increased open waste burning as well reduced wet scavenging and ventilation coefficient compared to the monsoon season. We examine these in more detail in the next sections.

**3.2: Time series of VOC tracers during the "clean" monsoon and "polluted post-monsoon" seasons in Delhi**

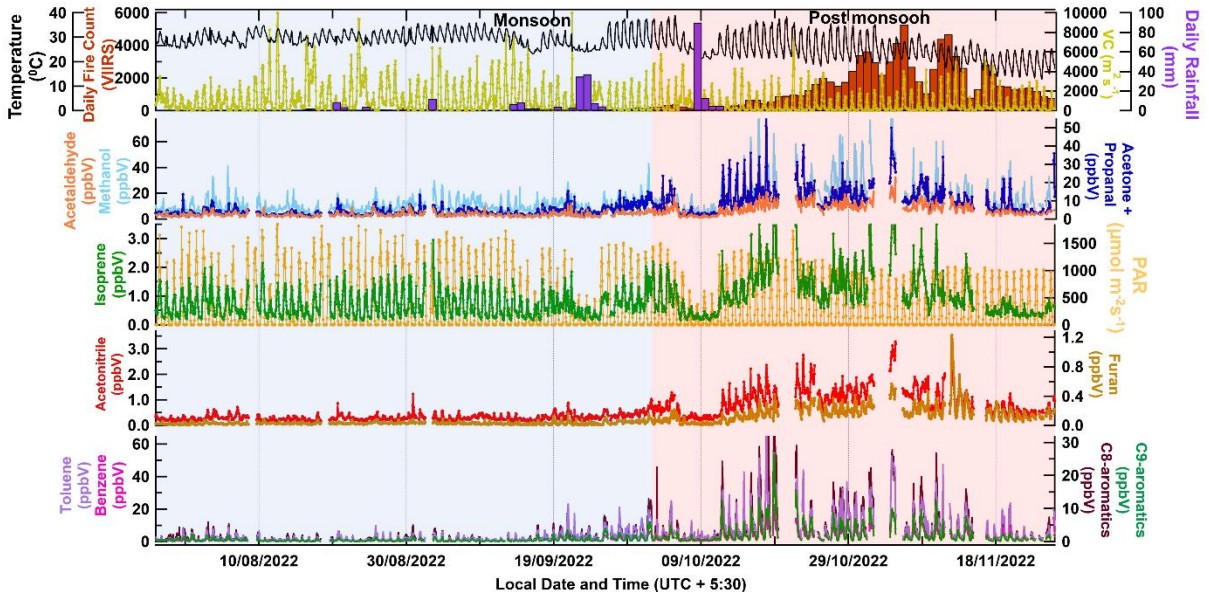



**Figure 3: Time series of hourly data for meteorological parameters like temperature (C) and ventilation coefficient**
**(m²s⁻¹), daily rainfall and daily fire counts (top panel); hourly mixing ratios of methanol, acetaldehyde, and the sum**
**of acetone and propanol (second panel from top); isoprene and PAR (µmolm⁻²s⁻¹) (third panel); acetonitrile and furan**
**(second panel from bottom); and benzene, toluene and the sum of C8 – aromatics (xylene and ethylbenzene isomers)**
**and the sum of C9 – aromatics (isomers of trimethyl benzene and propyl benzene) (bottom panel). The blue and red**
**shaded regions represent the monsoon and post-monsoon periods, respectively.**

Figure 3 shows the time series plot of meteorological parameters and the mixing ratios of some key VOC tracer
molecules during monsoon (22$^{nd}$ July – 30$^{th}$ September 2022, blue-shaded region) and post-monsoon (1$^{st}$ October
– 26$^{th}$ November 2022, red-shaded region). The top panel shows the ambient Temperature ($^0$C), daily VIIRS fire
counts on the left side of the top panel and ventilation coefficient (m²s⁻¹), and daily rainfall (mm) on the right side
of the top panel during the study period (22$^{nd}$ July 2022 – 26$^{th}$ November 2022). A grid (1km × 1km) with latitudes
between 21$^0$N and 32$^0$N and longitudes between 78$^0$E and 88$^0$E was considered for extracting the fire count data.
The second panel from the top represents the time series of mixing ratios of OVOCs which can be formed photo-
chemically as well as be emitted from anthropogenic sources, namely methanol, acetaldehyde, and the sum of
acetone and propanol; the third panel shows the mixing ratio of isoprene (a daytime biogenic chemical tracer, pure
hydrocarbon) and photosynthetic active radiation (PAR) (µmol photons m⁻² s⁻¹), and the fourth panel shows the
mixing ratio of acetonitrile (a biomass burning chemical tracer) and furan (a combustion chemical tracer). The
bottom panel shows the mixing ratios of benzene, toluene, the sum of C8–aromatics (xylene and ethylbenzene
isomers), and the sum of C9–aromatics (trimethylbenzene and propyl benzene isomers). These are some of the
most abundant VOCs typically present in any urban megacity environment, due to their strong emission from
traffic and industries in addition to biomass burning (Sarkar et al., 2016; Sinha et al., 2014; Chandra et al., 2016;
Singh et al., 2023; Dolgorouky et al., 2012; Yoshino et al; 2012; Langford et al., 2010). We note that all the
meteorological conditions and fire activity and VOC levels changed significantly between the much "cleaner"
monsoon season and "highly polluted" post-monsoon season at the same site. While the average temperature
during monsoon season was 29.5±2.8 °C, in the post-monsoon season this changed to 24.8±5.2 °C, while the
average ventilation co-efficient was 1.7 times higher during monsoon season relative to the post-monsoon season.
Except for the period impacted by heavy rainfall due to western disturbance weather (8th Oct – 10th Oct 2022),
the average mixing ratios for all compounds were considerably higher in the post-monsoon season relative to the
monsoon season even after accounting for the ventilation coefficient reduction with all the aromatics compounds
like benzene, toluene, sum of C8 and C9 aromatics, all 4.5 times higher and furan more than 5 times higher and
acetonitrile, acetone more than 3 times higher and methanol and acetaldehyde 2 times higher. Even isoprene was
1.7 times higher but its night time mixing ratios were higher than daytime mixing ratios during post-monsoon
season relative to the monsoon season. The increases clearly exceed what can be accounted for only by the reduced
ventilation co-efficient (seasonality) and suggests an increase in anthropogenic combustion related sources in
particular from open biomass burning fire sources, which we investigate in more detail in the subsequent sections.
**3.3: Analyses of the diel profiles during the "clean" monsoon and "polluted post-monsoon" seasons in Delhi**
**for discerning major drivers of their ambient values**
Figure 4 shows the diel box and whiskers plot depicting the average, median, and variability (10th, 25th, 75th and
90th percentile) of the same key VOCs like methanol, acetonitrile, acetaldehyde, acetone and propanal, furan,
isoprene, benzene, toluene and C8 - aromatics for monsoon (derived from ~ 1704 data points, blue markers) and
post-monsoon (derived from ~1368 data points, red markers) against the hour of the day (the horizontal axis
represents the start time of the corresponding hourly bin). This more clearly brings out the season-wise diel
variation of the compounds and in turn throws light on the emission characteristics and how they vary for the
same compound between seasons. Both in the monsoon and post-monsoon season, methanol mixing ratios seem
to be driven by primary emission sources and correlate very well with toluene, a tracer for traffic emissions, with
highest increases in the evening hours (17:00 to 20:00 L.T.). Globally the main source of methanol is vegetation
but in a megacity like Delhi that possesses more than 150000 compressed natural gas (CNG) vehicles and light
duty diesel vehicles, it appears that traffic (see Fig 1 of (Hakkim et al., 2021) emitted methanol controls its ambient
abundance.  Similarly, based on the correlation with toluene, traffic emissions seem to be a major contributor for
acetaldehyde, acetone, sum of C8-aromatics and benzene in the morning and evening hours. All these compounds
are among the most abundant VOCs detected in tailpipe exhaust samples (Hakkim et al., 2021). Average ambient
mixing ratios of acetonitrile, a compound emitted significantly from biomass burning (Holzinger et al., 1999),
were below 0.5 ppb in the monsoon for all hours, with only slight increase at night, but during post-monsoon
season, for all hours the values doubled to 1 ppb, with strong increases in the early evening and night time hours.
This tendency was mirrored in all the other compounds including isoprene. The diel profile of isoprene and
acetaldehyde were the only ones which showed daytime maxima during the monsoon season.
This shows that during the monsoon season, the biogenic sources of isoprene majorly drive its ambient mixing
ratios, whereas acetaldehyde ambient mixing ratios are controlled by photochemical production of the compound
in the monsoon season. Under the high NOx conditions prevalent in a megacity like Delhi, photo-oxidation of n-
butane, propene, ethane and propane could be a large photochemical source of acetaldehyde (Millet et al., 2010).

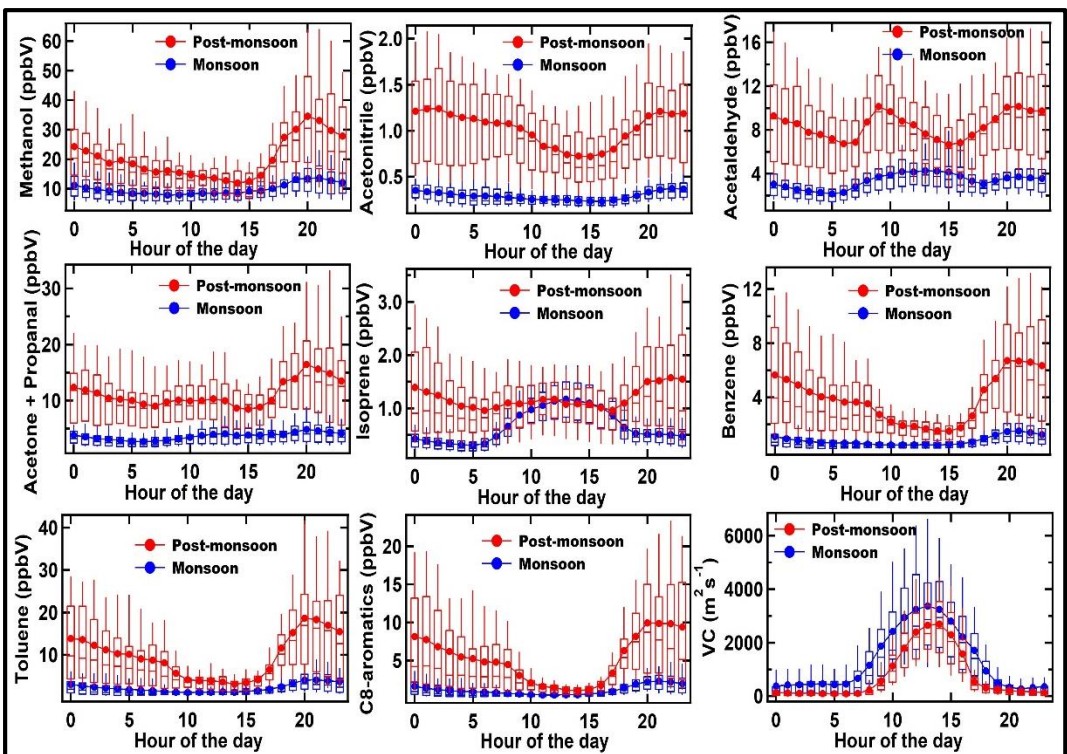

**Figure 4: Box and whisker plots showing average, median, and variability (10th, 25th, 75th and 90th percentile) for some major VOCs and the ventilation coefficients (m²s⁻¹) (VC) during monsoon and post-monsoon periods. The blue and red markers represent the monsoon and post-monsoon periods, respectively.**

Benzene which is human carcinogen is the only VOC for which there is a national ambient air quality standard (5 µg m⁻³ equivalent to ~1.6 ppb at 298 K) in India. Average mixing ratios in the post-monsoon season (Fig 4) were always above this value no matter what hour of the day, and the seasonal average was twice as high as this value (~4 ppb). The increased biomass burning in post-monsoon season controlled the abundance of benzene, acetaldehyde and acetone and isoprene during this period, due to strong emissions from both biomass burning and traffic. The typical atmospheric lifetimes of all these compounds spans from few hours (e.g. isoprene) to several days (e.g. benzene and methanol) and several months in the case of acetonitrile. The results of the TD-GC-FID measurements along with the average PTR-TOF-MS values presented in Figure 4, are summarized in Figure S7. Even though the TD-GC-FID measurements present only a snapshot as the ambient sampling duration is shorter, the season-wise diel profiles are consistent with those obtained using the PTR-TOF-MS and the average mixing ratios obtained using the PTR-TOF-MS dataset also fall well within the range of mixing ratios observed using the TD-GC-FID. This provides further confidence in the high night-time isoprene observed during the post-monsoon season. The isoprene emissions at night during the post-monsoon season are likely due to combustion sources. Paddy residue burning and dung burning have the highest isoprene emission factors of ~0.2 g/kg (Andreae 2019) and more than 8 Gg of isoprene is released in the space of a few weeks during the post-monsoon season regionally from open paddy residue burning alone (Kumar et al., 2021). Previous studies from the region have also documented isoprene emissions from non-biogenic sources, which are active also at night (Kumar et al., 2020, Hakkim et al., 2021). In 2018 at another site in Delhi, using gas chromatography measurements made in pre-monsoon and post-monsoon, Bryant et al. 2023 reported average nocturnal mixing ratios of isoprene that were 5 times higher in the post-monsoon compared to the pre-monsoon and showed different diel profiles between the

seasons. They found that the high night-time isoprene correlated well with carbon monoxide, a combustion tracer
and suspected that in addition to the stagnant meteorological conditions, biomass burning sources could be a
reason for the significant night time isoprene in Delhi in post-monsoon season and our findings using more
comprehensive and high temporal resolution data further substantiate the surprising night-time isoprene.
As potent precursors of secondary organic aerosol, the aromatic compounds would also enhance secondary
organic aerosol pollutant formation during the polluted post-monsoon season. When compared with the first PTR-
MS measurements of these compounds reported from wintertime Delhi (see Fig 2 of Hakkim et al., 2019), the
average levels of these compounds for the post-monsoon season (Table S2) are lower or comparable, but still
significantly higher than what has been reported for other major cities of the world like Tokyo, Paris, Kathmandu,
Beijing, London (Yoshino et al., 2012; Dolgorouky et al., 2012; Sarkar et al., 2016; Li et al., 2019; Langford et
al., 2010). The monsoon levels on the other hand were comparable to many of the other megacities.
As the monsoon season is characterized by favourable meteorological conditions for wet scavenging and dispersal
due to higher ventilation co-efficient, as well as significantly lower open biomass burning due to wet and warm
conditions, the monsoon levels can be considered as baseline values for the ambient levels of these compounds
(except isoprene and acetaldehyde) in Delhi, which are driven mainly by year-round traffic and industrial sources
in Delhi. In monsoon for isoprene, the major driver are biogenic sources whereas for acetaldehyde the major driver
is photochemistry, a finding that is similar to what has been reported from another site in the Indo-Gangetic Plain
previously (Mishra and Sinha, 2020).
**3.4: Discovery of methanethiol (CH$_3$SH), dichlorobenzenes (C$_6$H$_4$Cl$_2$), and C6-amides (C$_6$H$_{13}$NO$_2$) and C9-**
**organic acids (C$_9$H$_{18}$O$_2$) in ambient Delhi air**
Figure 5 shows the average diel profile of four compounds present in both monsoon and post-monsoon periods
that have to our knowledge never been reported from Delhi or any site in South Asia and only rarely been reported
in the gas phase in any atmospheric environment in the world.  Except for methanethiol detected at m/z 49.007
(also called methyl mercaptan), all the other compounds namely dichlorobenzene (C$_6$H$_4$Cl$_2$) detected at at m/z
146.977, C6-amides like hexanamide (C$_6$H$_{13}$NO$_2$) and its isomers detected at m/z 116.108 and C9- carboxylic
acid/ester such as nonanoic acid (C$_9$H$_{18}$O$_2$) and its isomers detected at m/z 159.14, are all intermediate volatility
range organic compounds. The saturation mass concentration (C$_0$) of methanethiol, C6-amide, dichlorobenzene,
and C9 organic acid, were calculated using the method described in Li et al. 2016 using the following equation:
$C_0 = \frac{M\ 10^6\ p_0}{760\ R\ T}$                                                                      (1)
wherein M is the molar mass [g mol$^{-1}$], R is the ideal gas constant [8.205 x 10$^{-5}$ atm K$^{-1}$ mol$^{-1}$ m$^3$], p$_0$ is the
saturation vapor pressure [mm Hg], and T is the temperature (K). Organic compounds with C$_0$ > 3 x 10$^6$ µg m$^{-3}$
are classified as VOCs while compounds with 300 < C$_0$ < 3 x 10$^6$ µg m$^{-3}$ as Intermediate VOCs (IVOCs).
The presence of such reactive organic sulphur, chlorine and nitrogen containing compounds in the gas phase
provides new insights concerning the chemical composition and secondary chemistry occurring in air, during the
extremely high pollution events. Below we examine the sources and chemistry of these compounds in further
detail.
The diel profiles of both methanethiol and dichlorobenzene in both the monsoon and post-monsoon seasons were
similar (bimodal with afternoon minima), and controlled by the ventilation coefficient diel variability (see Fig. 4),
and in fact even the difference in their average magnitudes (50 ppt Vs 130 ppt for CH$_3$SH and 25 ppt Vs 100 ppt
for dichlorobenzene between monsoon and post-monsoon seasons), can largely be explained by the reduction in
ventilation co-efficient (~2 reduction). Further, the conditional probability wind rose plots for both compounds
shows that the high values come from the same wind sector upwind of the site spanning north-east to south during
early morning and evening hours, which is actually where a variety of industrial sources are located. Previously,
Nunes et al., (2005) and Kim et al., (2006) have reported methanethiol from petrochemical industries and landfills
in Brazil and Korea, respectively. Toda et al., (2010) reported high (tens of ppb) methanethiol mixing ratios from
a pulp and paper mill industry in Russia.

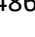

**Figure 5: Average diurnal profile of methanethiol, isomers of dichlorobenzene, C6-amides, and C9- organic acid in**
**the left panel for both monsoon (blue marker) and post-monsoon (red marker) periods. The second panel shows the**
**wind rose plot of methanethiol and isomers of dichlorobenzene, plot of C6-amide vs ozone and C9-organic acid vs**
**RH$^{-1}$ colour coded by the hour of the day. The third panel shows the polar annulus plot of methanethiol, isomers of**
**dichlorobenzene and wind rose plot of C6-amides and C9-organic acid.**
Both compounds are also used in the deodorant and pesticide products as reagents (Chin et al., 2013) and although
large scale pesticide manufacturing facilities were shifted out of Delhi, there are still units that sell and distribute
these products in those areas, from which fugitive emissions are likely happening. Methanethiol is further used as
a precursor in methionine production (Francois, 2023) an essential amino acid used in manufacture of pesticides,
and fragrances industry uses methanethiol for its distinct sulphur-like aroma (Bentley et al., 2004), contributing
to the creation of savory flavors and unique fragrances. In the Delhi environment, a combination of such industries,
in particular paper and pulp industries, are likely candidate sources. Figure 5 confirms that elevated methanethiol
values in the windrose had a clear directional dependence from the area spanning the north to south east sector.
This is the region where various manufacturing facilities and industrial areas of Delhi like Patparganj (north-east)
Okhla, Faridabad (south) are situated and these industrial estates were earlier marked in Figure 1 b. Methanethiol
is an extremely reactive molecule reacting primarily with the hydroxyl radical (OH) during daytime with an
estimated lifetime of 4.3 h (Wine et al., 1981). Its photo-oxidation in daytime with hydroxyl radicals produces
sulphur dioxide, methanesulfonic acid, dimethyl disulphide and sulphuric acid (Kadota and Ishida et al., 1972;
Hatakeyama et al., 1983), all of which play key roles in aerosol formation pathways. Dimethyl disulphide has a
very short atmospheric lifetime spanning from 0.3 to 3 hours (Hearn et al., 1990), because of its high reactivity
($[1.98 \pm 0.18] \times 10^{-10}$ cm$^3$ molecule$^{-1}$s$^{-1}$) (Wine et al., 1981) with OH radicals. Although dimethydisulphide is the
major product of the photo-oxidation of methanethiol (yield 50%; Wine et al., 1981), since methanethiol itself
was on average only 48 ppt (monsoon) and 128 ppt (post-monsoon), and plumes occur only at night we
hypothesize that the ambient concentrations of DMDS were too low to be detected by the mass spectrometer.
Further it can also react with nitrate radicals (Berreshiem et al., 1995) and participate in night-time chemistry.
More recently, Reed et al. (2020) performed laboratory experiments and observed that even trace amounts of
organosulphur compounds, such as thiols and sulfides, can significantly enhance the organic aerosol mass
concentration and its particle effective density. Though there has not been any relevant data set attributing the
enhancement of organic aerosols to methanethiol in Delhi specifically, previous studies have found enhanced
secondary aerosol formation rates during haze and fog episodes (Acharja et al., 2022). These studies collectively
suggest an increase in the haze events in Delhi is linked to sulphur chemistry in which methanethiol due to its
high reactivity and atmospheric chemistry could also be a contributor along with ammonia and other sulphur
containing molecules. Wine et al. (1981) had further predicted that the very rapid rate at which methanethiol reacts
with OH would result in low steady-state concentrations in ambient air, even though reasonably large-scale
sources may exist.
Several recent studies have reported high chloride in sub-micron aerosol of Delhi (Gani et al., 2020; Acharja et
al., 2023; Pawar et al., 2023). Dichlorobenzene is an intermediate range volatile organic compound (IVOC) which
can partition between gas and aerosol phase. However, till date no gaseous IVOC chlorinated organic compound
have been reported in ambient air from India. p-dichlorobenzene (PDCB) also called 1. 4-dichlorobenzene, one
of the dichlorobenzene isomers is known for its use as a pest repellent and deodorant in indoor environments. 1,4-
dichlorobenzene in outdoor air in various locations of North America and Europe ranged from 30 ppt to 830 ppt
(Chin et al., 2013). It is emitted only from anthropogenic sources as there are no known natural sources. Its
emission sources include consumer and commercial products containing PDCB, waste sites, and manufacturing
facilities for flavour and as insect repellent products (ATSDR. 2006). Its atmospheric lifetime is estimated to be
21-45 days (Mackay et al., 1997).  It has been reported as a precursor of secondary organic aerosol in indoor
conditions (Komae et al., 2020). Due to its long lifetime, dichlorobenzene can be transported to upper regions of
the atmosphere where some release of some reactive chlorine through photolysis can occur, but this is not likely
to be of large consequence. Instead, reaction with hydroxyl radicals would convert it more readily to phenolic
compounds that would readily partition to aqueous aerosol phase and also undergo nitration to form nitrophenolics
(Hu et al., 2021), which are a component of brown carbon (Lin et al. 2015, 2017).
In contrast, the diel profile of the average mixing ratios of $C_6H_{13}NO$ (Fig. 5), likely hexanamide or isomers of C6-
amides measured at m/z 116.108, was similar in both monsoon and post-monsoon season and characteristic of a
compound with a purely photochemical source with no evening time peaks even during the enhanced biomass
burning in post-monsoon season. As observed for several other compounds in this study, the difference in
magnitude between both seasons (peak value 22 ppt in post-monsoon season vs 12 ppt in monsoon season) could
be accounted for almost completely by the reduced ventilation co-efficient in post-monsoon season (factor of ~2).
The presence of photochemically formed formamide and acetamide from OH oxidation of alkyl amine precursors
has been previously reported (Chandra et al., 2016; Kumar et al., 2018), from another site in the Indo-Gangetic
Plain which experiences strong agricultural waste burning. In the literature we could only find only one report for
presence of C6 amides in the ambient air in the gas phase (Yao et al., 2016), who reported ~14 ppt in summertime
air of Shanghai using an ethanol reagent ion CIMS, the source of which was both industrial and photochemical
origin. However, to our knowledge this is the first study world-wide to detect and report only photo-chemically
formed C6-amides in the gas phase. C6-amides are IVOCs, which can easily partition to aerosol phase depending
on environmental conditions and also act as a new source of reactive organic nitrogen to the atmospheric
environment. We found the highest values in air masses arriving in the afternoon from the north-west direction at
high wind speeds (see Fig 5) during the post-monsoon season, which indicated that paddy stubble burning
emissions of amines (Kumar et al., 2018) were its likely precursors. The mechanism of amide formation through
photochemical reactions has been elucidated in several previous laboratory studies (Bunkan et al., 2016, Barnes
et al., 2010; Nielsen et al., 2012; Borduas et al., 2015). When correlated with daytime ozone hourly mixing ratios,
the very high correlation ($r^2 > 0.97$), confirmed its purely photochemical origin. Being an amide, further gas phase
oxidation products are likely to result in organic acids or condensation on existing aerosol particles which could
add to the reactive organic nitrogen in aerosol phase and neutralize acidity just like ammonia, as ammonium ion
is formed from hydrolysis of amides (Yao et al., 2016). However, the exact role of these amides in nucleation and
aerosol chemistry will warrant further investigations.
Finally, the last row of Fig. 5 shows the average mixing ratios of the compound with molecular formula $C_9H_{18}O_2$
which is likely due to isomers of C9- carboxylic acids (e.g. nonanoic acid), although one cannot rule out
contributions from isomers of esters such as methyl octanoate or 2-methylbutyl isobutyrate also detected at m/z
159.14. Hartungen et al. (2004) and more recently the study by Salvador et al. (2022), have highlighted that
carboxylic acids (RCOOH) can undergo dissociation reactions within the drift tube in addition to protonation, and
form acylium ions as per the following reaction below (Hartungen et al., 2004):
$H_3O^+ + RCOOH \rightarrow RCOOHH^+$ (protonated ion) $+ H_2O$                                        (R1)
$H_3O^+ + RCOOH \rightarrow RCO^+$ (acylium ion) $+ 2 H_2O$                                        (R2)
We detected the corresponding acylium ion of C9-carboxylic acid ($C_8H_{17}CO^+$ detected at m/z 141.13) in the
measured ambient spectra (Figure S9) and found that not only was it present but that it also correlated well in the
ambient data with the protonated ion (r=0.83). The presence of the fragment ion and its correlation, provides
additional confirmation concerning the attribution of m/z 159.14 to the C9-organic acid and for quantification the

ion signals due to the protonated and acylium ions, were summed together for greater accuracy. Although here also there is a daytime peak, the timing of the peak is much later in the day (15:00 local time). The peak hourly values reached 60 ppt in post-monsoon season. It showed high correlation ($r^2 > 0.93$) with the inverse of the ambient daytime relative humidity indicating that it partitions back and forth between the gas phase and aerosol phase depending on the environmental conditions of temperature and RH. n-alkanoic acids in general and nonanoic acid in particular have long been reported as major organic acids present in biomass burning emitted organic aerosol (Oros et al., 2006; Fang et al., 1999). The corresponding wind rose plot (Fig. 5) shows that the highest values were in air masses arriving at high wind speeds in the afternoon from the north-west during post-monsoon season, which is a major source region of biomass burning emitted organic aerosols. It is also possible that photochemical oxidation through ozonolysis of precursors and hydroxyl radical initiated oxidation can form such carboxylic acids as an advanced oxidation product (Kawamura et al., 2013). In both cases, biomass burning emissions and evaporation from aerosol phase, appear to be the major source of this compound. Carboxylic acids in the aerosol phase would serve to neutralize some of the excess ammonia in the atmospheric environment of the Indo-Gangetic Plain (Acharja et al., 2022) and would be important for night-time aerosol chemistry in Delhi.

**3.5: Comparison of ambient mixing ratios and VOC/CO emission ratios for aromatic VOCs in Delhi with some megacities of Asia, Europe and North America**

Aromatic compounds are among the most important class of compounds in urban environments due to their direct health effects (e.g. benzene is a human carcinogen), and reactivity as ozone and secondary organic aerosol precursors. Therefore, these compounds have been widely investigated in many cities and information concerning their ambient levels and emission ratios to carbon monoxide is often used for assessing similarities and differences in the sources of these compounds in varied urban environments (Warneke et al., 2007' Borbon et. al., 2013). In Figure 6, we show the emission ratios (ER) derived for benzene, toluene and the sum of C8 and C9 aromatic compounds (VOC / CO ppb/ppm) using night-time monsoon (left panel) and post monsoon (right panel) measurements made in Delhi. The method is based on a linear regression fit to determine the slope of the night-time scatterplot data (from 20:00 to 06:00 L.T.) between a VOC (ppb) and CO (ppm) (de Gouw et. al., 2017, Borbon et. al., 2013). Using night-time hourly data (18:00 to 06:00 L.T.) provides the advantage of minimizing complications due to daytime oxidative losses of the compounds. It can be noted from Fig. 6, that during the monsoon season (from 18:00 to 23:00 and 00:00 to 06:00 local time) and post-monsoon season (18:00 to 23:00), the observed emission ratios as inferred from the slopes and fits are not statistically different from each other (all highlighted by oval circles) with values for benzene/CO, toluene/CO, sum of C8-aromatics, sum of C9-aromatics/CO in the range of 1.2-2.43, 3.14-6.76, 1.97-3.84, and 1.05-2.07, respectively. All these emission ratios fall within the range of what has been reported for typical petrol 2 and 4 wheeler vehicles in India in tail pipe emissions (Hakkim et al., 2021). For the monsoon season, although two linear fits are observed from 18:00 to 23:00 and 00:00 to 06:00, the values of the emission ratios as inferred from the respective slopes for all compounds overlap or are very close to each other and within the uncertainties for all compounds. We hypothesize that the two fits are due to the change in relative numbers of 2 wheelers and 4 wheelers. In the post-monsoon season however, for the time period in the second half of the night (00:00-06:00), the emission ratios derived from the slopes are statistically different from the ones observed in monsoon season and the first half of night in post-monsoon season (18:00-23:00). When we examined the wind rose plots for the same night-time data of the aforementioned compounds for each season (Figure S8), we noted that during the post-monsoon

season more pollution plumes from the south east sector which has industrial facilities and the north west sector (a major fetch region for biomass burning plumes from regional paddy residue burning in Punjab and Haryana) occurred. During the post-monsoon season due to dip in temperatures at night, the heating demand (Awasthi et al., 2024) and associated open biomass burning (Hakkim et al., 2019) also goes up, relative to the monsoon period nights. Hence overall we think that these additional sources in the post-monsoon season, do add to the burden of these mainly traffic emitted aromatic compounds and could help explain atleast partially the higher emission ratios observed during the post-monsoon season ( 00:00- 06:00), wherein values for benzene/CO, toluene/CO, sum of C8-aromatics , sum of C9-aromatics/CO values range from 3.15-3.27, 7.72-8.68, 5.03-5.37, 2.6-2.76, respectively, and are statistically different from the others (ones marked by oval circles).

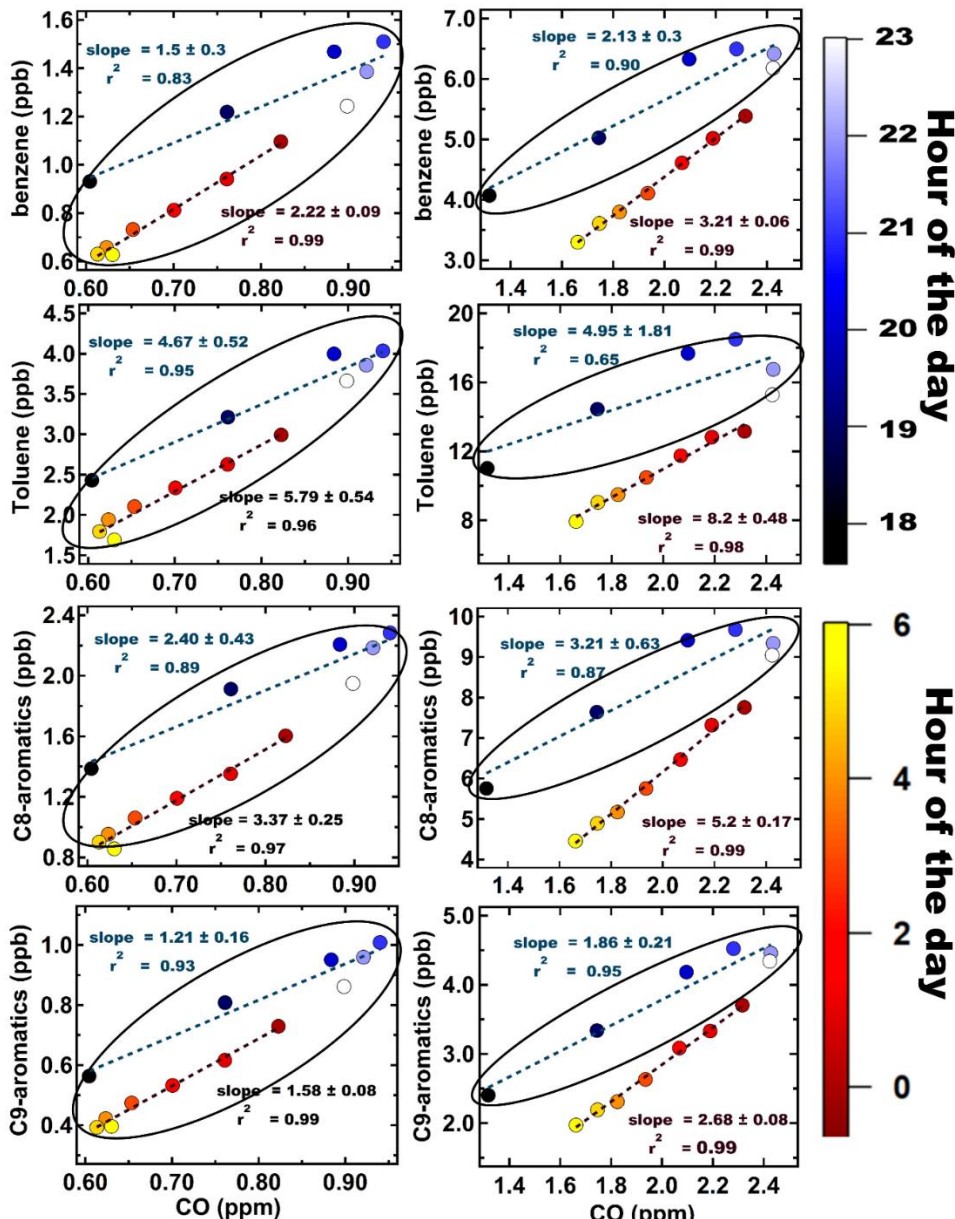

**Figure 6: Emission ratios (VOC (ppb)/CO (ppm)) of benzene, toluene, C8 aromatics and C9 aromatics for both monsoon (left panel) and post monsoon (right panel) periods respectively. The data points for each period are colour coded with the hour of the day (18:00 L.T to 06:00 L.T).**


Table 1 provides a comparison of the ambient mixing ratios and emission ratios that have been reported in some
other major megacities of Asia, Europe and North America for these compounds. Although, the year of
measurements and seasons are not the same, nonetheless such comparison helps put the 2022 levels of these
compounds in Delhi in a global context. It may be further noted that we took care to calculate the emission ratios
using only night-time data when chemical loss of these compounds is negligible as their main oxidation is through
OH radicals during daytime, as also noted by de Gouw et al., 2017. Further, the other studies referred to in Table
1 for comparison, have also reported emission ratios derived using only nighttime data.

**Table 1: Comparative summary of the average mixing ratio (ppb) and Emission Ratios of VOC/ CO (ppb/ppm) of**
**Delhi (in parentheses) with other megacities of Asia, Europe and North America**

| VOC | Delhi* | Langzhou Valley[1] | Sao Paulo[2] | London[3] | Los Angeles[4(a)] | Paris[5(a)] | Mexico City[6(b)] | New York[4(c)] | Beijing[7(d)] | Lahore[8] |
|---|---|---|---|---|---|---|---|---|---|---|
| **Benzene** | 2.02 | 0.54 | 0.67 | 0.31 | 0.48 | 0.38 | 0.80 | 0.74 | 1.79 | 28.20 |
| | (2.65) | (1.37) | (1.03) | (1.59) | (1.30) | (1.07) | (1.21) | (1.09) | (1.24) | (5.08) |
| **Toluene** | 5.15 | 0.72 | 2.11 | 0.60 | 1.38 | 1.40 | 3.10 | 0.19 | 1.98 | 32.40 |
| | (7.03) | (1.41) | (3.1) | (3.09) | (3.18) | (12.30) | (4.20) | (3.79) | (2.41) | (6.67) |
| **Sum of C8 aromatics** | 2.74 | 0.61 | 1.52 | 0.63 | 1.03 | 1.30 | 1.10 | 0.88 | 2.66 | 29.40 |
| | (4.20) | (1.42) | (2.15) | (3.69) | (2.45) | (4.75) | (4.30) | (1.11) | (2.15) | (6.04) |

*This work (2022)   [1]Zhou et al., (2019)   [2]Brito et al., (2015)   [3]Valach et al., (2014)   [4]Baker et al., (2008)
(a)Borbon et al., (2013)  [5]Gros et al., (2011)   [6]Garzón et al., (2015)   [7]Yang et al., (2019a)   [8]Barletta et al.,
(2016) (b)Bon et al., (2011)   (c)Warneke et al., (2007) (d)Wang et al., (2014)   +Apel et al., (2010)

Except for Lahore, where benzene levels were 10 times higher, benzene levels in Delhi were comparable to Beijing
and about three times higher than those that have been reported from other megacities like Sao Paulo, London,
Los Angeles, Paris, Mexico City and New York. The annual averaged national ambient air quality standard for
benzene is 5 $\mu g$ $m^{-3}$ in India which is approximately 1.6 ppb at room temperature. Thus, the data suggest that
sources in the investigated period (Monsoon and Post-monsoon season) would contribute to violation of the annual
averaged values. Similarly, toluene and the sum of C8 aromatic compounds (e.g. xylene and ethyl benzene
isomers) were 6 to 10 times higher in Lahore compared to Delhi and more than twice as high relative to the
aforementioned megacities, except for Beijing, where the sum of C8 aromatic compounds were comparable to
Delhi. Overall, this indicates that Delhi has much higher levels of aromatic VOC pollution than many other
megacities. When we peruse the emission ratios (ER) that have been reported for these compounds in these other
megacities (shown in parentheses in Table 1), barring few exceptions (e.g. Lahore and Paris), the ERs were
generally much higher in Delhi with an average value of 2.65, as compared to cities like Sao Paulo (Brito et al.,
2015), London (Valach et al., 2014) and Los Angeles and Paris (Borbon et al., 2013), Mexico City (Bon et al.,
2011) and several US cities (Baker et al., 2008). The ER of toluene was highest in Paris (12.3) followed by Delhi.
Overall, the mixing ratios and ERs indicate that the influence of non –traffic sources (e.g. biomass burning and
industries) is more significant in Delhi compared to many other megacities of the world. The companion paper on
source apportionment based on this dataset (Awasthi et al., 2024) will focus more on the quantitative contributions
of the different sources.

**4. Conclusion**

This study has provided unprecedented characterization of the VOC chemical composition of ambient air in Delhi for the clean monsoon and extremely polluted post-monsoon seasons. The total average mass concentration of the reactive carbon in the form of the 111 VOC species identified unambiguously was ~260 $\mu gm^{-3}$ and more than 4 times higher during the polluted post-monsoon season mainly due to the impact of large scale open fires and reduced ventilation relative to the "cleaner" monsoon season. Of the 111, 42 were pure hydrocarbons (CH), 56 were oxygenated volatile organic compounds (OVOCs; CHO), 10 were nitrogen containing compounds (NVOCs; CHON), 2 were chlorinated volatile organic compounds (ClVOCs), and 1 namely methanethiol, contained sulphur. The detection of new compounds that have previously not been observed in Delhi's air, under both the clean and polluted periods such as methanethiol, dichlorobenzenes, C6-amides and C9-organic acids in the gas phase was very surprising, considering there have been several PTR-TOF MS studies in Delhi earlier (Wang et al., 2020; Tripathi et al., 2022; Jain et al., 2022). Our data points to both industrial sources of the sulphur and chlorine compounds, photochemical source of the C6-amides and multiphase oxidation and chemical partitioning for the C9-organic acids. To our knowledge this is the first reported study world-wide to detect and observe only photo-chemically formed C6-amides in the gas phase. C6-amides are IVOCs, which can easily partition to aerosol phase depending on environmental conditions and also act as a new source of reactive organic nitrogen to the atmospheric environment.

The monsoon season VOC abundances for major compounds were comparable to several other megacities of the world showing that the baseline VOC levels for the city of Delhi due to year-round active sources, helped by favourable meteorological conditions for removal of VOCs through ventilation and wet scavenging, can lead to comparable air quality as observed in other megacities. The VOC levels during the polluted post-monsoon season when severe air pollution events occur leading to shutdowns and curbs, on the other hand were significantly (2-3 times) higher. Overall, for many important aromatic VOCs, the levels measured in Delhi were even higher (> 5 times) than many other megacities of the world located in Europe and North America. Generally these aromatic compounds in megacities are primarily due to traffic and industrial emission sources, and this source is of course common to Delhi and megacities in Europe and North America. In Delhi, the highest ambient mixing ratios of these aromatic compounds occurred in the post-monsoon season. This is the period when enhanced open biomass burning occurs due to heating demand increase owing to dip in temperatures (Hakkim et al., 2019; Awasthi et al., 2024) and open fire emissions due to the seasonal post-harvest paddy stubble biomass burning in which more than 1 billion ton of biomass is burnt regionally (Kumar et al., 2021) within few weeks during mid-October to end of November occur. This adds significantly to the atmospheric burden of these compounds, compared to megacities in developed countries where open biomass burning is better and more strictly regulated. Secondly, the meteorological conditions during post-monsoon season due to shallower boundary layer height and poor ventilation, and lack of wet scavenging due to absence of rain also slow down atmospheric removal of these compounds compared to megacities in Europe, wherein it rains more frequently throughout the year compared to Delhi.

The presence of such a complex mixture of reactant VOCs adds to the air pollutant burden through secondary pollutant formation of aerosols. The reactive gaseous organics, which reached total averaged mass concentrations of ~85 $\mu gm^{-3}$ (monsoon season) and ~265 $\mu gm^{-3}$ (post-monsoon season) were found to rival the high mass concentrations of the main air pollutant in exceedance at this time, namely $PM_{2.5}$ during the extremely polluted

periods (post-monsoon season average: ~145 $\mu gm^{-3}$ which exceeds the 24h national ambient air quality standard
of 60 $\mu gm^{-3}$). The data of the time series of the $PM_{2.5}$ hourly data along with acetonitrile (a biomass burning VOC
tracer) measured at the same site is provided in Figure S10. While the present study has quantified the molecules
in the gas phase that are important for the air chemistry driving the high pollution events in Delhi in unprecedented
detail, the implications on secondary pollutant formation will require building up on this new strategic knowledge
and further investigations. Moreover, the unique primary observations will yield quantitative source
apportionment of particulate matter and VOCs in a companion study (Awasthi et al., 2024), that is being been co-
submitted to this journal to enrich the scientific insights.
All previous VOC studies in the literature from a dynamically growing and changing megacity like Delhi were
reported for periods before 2020 (pre-COVID) times, without the new enhanced volatility VOC quantification
technology deployed for the first time in a complex ambient environment of a developing world megacity like
Delhi. These have resulted in unprecedented new information concerning the speciation, abundance, ambient
variability and emission characteristics of several rarely measured/reported VOCs. The significance of the new
understanding concerning atmospheric composition and chemistry of highly polluted urban atmospheric
environments gained from this study, will no doubt be of global relevance as they would aid atmospheric chemistry
investigations in many megacities and polluted urban environments of the global south, that are in similar
development and growth trajectory as Delhi and experience extreme air pollution and air quality associated
challenges, but remain understudied.
**Data availability**
The primary VOC, CO and Ozone and meteorological data presented in this manuscript can be downloaded by
accessing the following Mendeley doi link: https://data.mendeley.com/preview/pb6xs2fzwc?a=7658dfde-2ca0-
46c8-b89b-54ba8211e1de

**Author Contribution**
Sachin Mishra: Data curation, Formal analysis, Investigation, Software, Visualization, Writing – original draft
preparation. Vinayak Sinha: Conceptualization, Data curation, Formal analysis, Methodology, Project
administration, Software, Supervision, Validation, Writing – review & editing. Haseeb Hakkim: Data curation,
Formal analysis, Investigation, Writing – review & editing.  Arpit Awasthi : Data curation, Formal analysis,
Investigation. Sachin D. Ghude: Writing – review & editing. Vijay Kumar Soni: Writing – review & editing. N.
Nigam: resources. Baerbel Sinha: Conceptualization, Data curation, Supervision, Writing – review & editing.  M.
Rajeevan: Writing – review & editing.
**Competing Interests**
The authors declare that they have no conflict of interest.

**Acknowledgment**

We acknowledge the financial support given by the Ministry of Earth Sciences (MOES), Government of India, to support the RASAGAM (Realtime Ambient Source Apportionment of Gases and Aerosol for Mitigation) project at IISER Mohali vide grant MOES/16/06/2018-RDEAS Dt. 22.6.2021. S.M acknowledges IISER Mohali for Institute PhD fellowship. AA acknowledges MoE for PMRF PhD fellowship. We thank Dr. R. Mahesh, Dr. Gopal Iyengar, Dr. R. Krishnan (Director, IITM Pune), Prof. Gowrishankar (Director, IISER Mohali), Dr. Mohanty (DG, IMD), Dr. M. Ravichandran (Secretary Ministry of Earth Science) for their encouragement and support. We thank student members of the Atmospheric Chemistry and Emissions (ACE) research group and Aerosol Research Group (ARG) of IISER Mohali and IITM Pune in particular Akash Vispute, Prasanna Lonkar and local scientists of IMD for their logistics and moral support. The authors gratefully acknowledge the NASA/ NOAA Suomi National Polar-orbiting Partnership (Suomi NPP) and NOAA-20 satellites VIIRS fire count data used in this publication. The authors gratefully acknowledge the NOAA Air Resources Laboratory (ARL) for the provision of the HYSPLIT transport and dispersion model used in this publication. We thank Campbell Scientific India Pvt Ltd, Ionicon Analytic GmbH and Mars Bioanalytical for technical assistance rendered by them.

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
