# Peer review of "Reactive chlorine, sulphur and nitrogen containing volatile 1"

_EGUsphere, 2024_

## Author Comment (AC1)

**Discovery of reactive chlorine, sulphur and nitrogen containing ambient volatile organic compounds in the megacity of Delhi during both clean and extremely polluted seasons**

**Sachin Mishra et al. 2024**

We express our gratitude to Reviewer 1 for her/his careful and detailed reading of the manuscript and the insightful and helpful comments, which we were pleased to peruse. The helpful comments have contributed to improving the context, messaging and clarity of the original submission. Below, please find point-wise replies to the reviewer's comments and suggestions.

The point-wise replies (in blue) to the reviewer's comments (**in black**) are below. Changes made to the text are also provided after the replies (in red).

**Response to Anonymous referee #1**

Sachin Mishra et al. report VOC measurements in the megacity Delhi, where they applied a high resolution PTR-ToF-MS for the first time, which allowed them to observe species that had so far not been reported from there. Since VOC observations from India and South Asia in general are scarce, this work is a valuable and important addition to the literature. However, I have some concerns that should be addressed before publishing the paper in ACP.

We thank the reviewer for deeming the submitted work to be a valuable and important addition to the literature and below we make every effort to address the valid concerns raised by the esteemed reviewer.

General comments:

The paper is well structured and readable. However, in some places, e.g. in the abstract or at the end of the introduction, the paper reads a bit like an advert for the company selling the PTR-ToF-10k. I would ask the authors to cut down on the advertising. E.g., it is not necessary to include the brand name ("10k") in the abstract. This is only necessary in the method description.

Thank you for deeming the paper well structured and easy to read! We regret the impression caused to the reviewer as our intention as fellow practitioners of the PTR-MS technique was simply to distinguish the instrument used in this work from previous PTR-MS deployments in Delhi, for which highlighting the model number-10K was useful to emphasize the significantly better mass resolution and sensitivity over previous instrument models, including by the same manufacturer, that were deployed in South Asia in previous studies. Appreciating the reviewer's point, we have removed mention of -10K from the abstract and few other places where it was repetitive but retained it the methods section.

I have a general concern about the identification of compounds that the authors claim to be "unambiguous". For example, they claim to have observed 42 pure hydrocarbons. But nowhere do they mention that the PTR method is subject to fragmentation and that these hydrocarbon masses could easily be, in part, fragmentation products of higher masses (Coggon et al., 2024; Pagonis et al., 2019).

We regret the confusion. Our use of "unambiguous" in no way implied that fragmentation isn't an important issue to consider, especially in a complex emission environment like Delhi. By "unambiguous" we simply implied that the attribution of the ion to its exact elemental composition as represented by its molecular formula, in terms of C, H , O , N and S atoms is unambiguous. This is in line with the usage of the term in the PTR-MS community previously as well where the term "unambiguous" was used to emphasize that in PTR-MS systems with lower mass resolution such as the PTR-TOF-MS 8000 (which in reality has mass resolution of 5000 for much of its dynamic range), even with similar mass accuracy, multiple peaks can be erroneously considered as a single one and remain unresolved as illustrated in the Figure shown below.

[Figure]

Image source: https://www.ionicon.com/blog/2023/the-advantages-of-high-mass-resolution-ptr-tofms-in-food-flavor-science

To quote from the description about the above Figure: "the superiority of the PTR-TOF 10k was found during the analysis of the complex headspace of chocolate for the presence of ethyl maltol ($C_7H_8O_3.H+$, m/z 141.055). With 5,000 m/$\Delta$m mass resolution at nominal m/z 141 the presence of five mass peaks is indicated (grey line). Interestingly, at about m/z 141.05 there appears to be rather a dip than a peak. By looking at the PTR-TOF 10k data (blue line) acquired from the same sample, the situation gets immediately **unambiguous**."

Further due to the enhanced sensitivity of the PTR-TOF-10K used in our work (even reaching <1 ppt for some compounds), it was also possible to identify the corresponding major isotopic peaks.

Thus by additional manual checks that the height of the isotopic peak was not different from the expected peak height (e.g. for isoprene $C_5H_8$, the contribution to C-13 isotopes would be expected to be

~5.5%), we could further confirm that the parent peak's molecular assignment was correct and also not attribute the shoulder isotopic peaks to another compound with different elemental composition, as in that case, the height of the isotopic peak would differ from the value expected based on the natural abundance of isotopes of carbon, hydrogen, sulphur, nitrogen, chlorine and oxygen.

To illustrate this further, we call attention to data reported in a previously published paper from data acquired in India that used a PTR-TOF-8000 in the Table below. It can be noted that the C-13 isotopic contribution of a compound like furan with m/z 70.037 was likely mis-assigned to triazole, or at very least contribution of the isotope of furan which is more abundant was not subtracted as far as can be made out from the details provided in that work. Similarly, the C-13 isotope of isoprene detected at m/z 70.073 would be more plausible for the observed peak at m/z 70.073, which was attributed to butane nitrile by the authors. In our assignment due to the higher sensitivity and mass resolution of our PTR-TOF-MS-10K , we were able to detect the isotopic peaks and ensure those were not misassigned.

Data below sourced from Supplementary Table S1 (Jain et al., 2023, Atmos. Chem. Phys., 23, 3383–3408, 2023, https://doi.org/10.5194/acp-23-3383-2023-supplement )

| Reported m/z | Chemical formula assigned | Species name | Theoretical m/z | Mass accuracy (Da) |
|---|---|---|---|---|
| 70.038 | $C_2H_3N_3$ | Triazole | 70.041 | -0.003 |
| **More plausible candidate compound** | **$C_4H_4O$** | **Isotope of furan** | **70.037** | **0.001** |
| 70.073 | $C_4H_7N$ | Butane nitrile | 70.066 | 0.007 |
| **More plausible candidate compound** | **$C_5H_9$** | **Isotope of isoprene** | **70.074** | **-0.001** |

*The bold ones are the best fit at the reported m/z*

It can be noted from the above that:

1.      A species named triazole ($C_2H_3N_3$) was assigned at m/z 70.038 considering a mass accuracy error of -0.003 Da, but the isotope of furan would seem to be the correct one at this m/z as the mass accuracy error for the latter  is only 0.001 Da. Note that isoptopic peaks for the two would be different too as the elemental composition is different with one containing nitrogen and the other not.

2.      A species named butane nitrile ($C_4H_7N$) was assigned at m/z 70.073 considering a mass accuracy of 0.007 Da, but the isotope of isoprene would be more plausible at this m/z as the mass accuracy error for this is only - 0.001 Da. Again the isotopic peaks for the two would be different too as the elemental composition is different with one containing nitrogen and the other not.

We hope the above clarifies what the context in which  the term "unambiguous" was used.

Appreciating that the meaning of this was not sufficiently clear in the original submission and also in response to a subsequent point mentioned concerning fragmentation at L199 later, we have now added the following text to make it more clear in the revised version after L199-200 ( ACPD version of pdf):

"The term "unambiguous" is used in the context of the accurate elemental composition/molecular formula assignment of the ions by leveraging the high mass resolution (8000-13000 over entire dynamic mass range) and detection sensitivity (reaching even 1 ppt or better for many ions; see Table S2) of the instrument. This enabled ensuring peaks due to expected isotopic signals were not construed as new compounds if their height was exactly as expected for a shoulder isotopic peak based on the natural abundance of isotopes of carbon, hydrogen, nitrogen, sulphur, chlorine and oxygen that made up the more abundant molecular ion. Where an ion could occur significantly due to fragmentation of another compound, the same has also been noted in Table S2 during attribution of the compound's name."

We agree wholeheartedly with the reviewer that fragmentation is an important issue that needs to be carefully considered while interpreting the mass assignments to specific compounds. We note that we considered known fragmentation issues and in Table S2 of the original submission, we also listed 8 of the 42 hydrocarbons that are likely to have significant contributions from fragment ions. This was based on the known fragmentations for the instrument operating conditions published previously in PTR-MS reviews, some of which (e.g. Yuan et al., 2017; Pagonis et al., 2019) we already cited in the original submission and is also mentioned in the reviewer's comment.

We thank the reviewer for drawing our attention to Coggon et al., 2024, which is a very valuable new study that sheds light on the impact of fragmentation on attribution of measured PTR-MS mass signals to important compounds like isoprene, toluene, benzene in atmospheric environments that are richly influenced by petrochemical industries and in specific urban environments. We also note that since our paper was submitted at the same time as Coggon et al. 2024 was finally published, we missed including it in our orginal submission and are grateful to the reviewer for pointing it out since including a discussion about its findings will help strengthen the new findings reported in the current study.

The suggestions and specific points are detailed below.

This includes isoprene, which can have substantial interference from higher aldehydes as well as substituted cyclohexanes (Coggon et al., 2024). Coggon et al. also described that the acetaldehyde mass is subject to interference from ethanol fragments, and benzene to fragments of other aromatics. I think the authors need to characterize their instrument's fragmentation at least for the most important compounds that they show diel plots for. The paper by Coggon et al. gives some pointers on how to correct for them, as well. Alternatively, if GC measurements were made at the same time, the authors could use those to identify compounds unambiguously.

We appreciate the reviewer's suggestions, especially since we also reported significant night time isoprene mixing ratios during the post-monsoon season, which may be surprising when compared to many ambient studies conducted in Europe and the USA, where isoprene is emitted primarily in the daytime from biogenic sources as a function of temperature and radiation. Besides considering the potential interferences due to fragmentation of higher molecular weight compounds as had already been flagged in previous PTR-MS reviews, we took additional measures to confirm the identity and diel profiles for the major compounds mentioned by the reviewer such as isoprene, benzene and toluene. Thus, during our campaign whole air samples were collected into 6 L passivated SilcoCan air sampling steel canisters (Restek) (Kumar et al., 2020; Vettikkat et al., 2020, Shabin et al., 2024) from the rooftop of the same building to measure isoprene, toluene and benzene using a thermal desorption gas chromatograph coupled to flame ionisation detectors (TD-GC-FID). Samples were collected during post monsoon season, four times a day [during start hourly intervals corresponding to 08:00, 13:00, 18:00, 20:00 local time; n=39] and during monsoon season also four times a day during the following hourly intervals [during start hourly intervals corresponding to: 00:00, 07:00, 12:00, 19:00 local time; n=15]. The results of the TD-GC-FID measurements in terms of season-wise hourly

averaged values and corresponding minimum and maximum values observed for the same hourly interval are shown in the graph below alongwith  PTR-TOF-MS data as reported in our original ACPD submission.

[Figure]

**New Figure S7:** Season-wise hourly averaged mixing ratios of isoprene, benzene and toluene measured using the TD-GC-FID are shown as diamonds and minimum and maximum values for that hourly sampling interval as shaded regions ( in green for isoprene, pink for benzene and purple for toluene). The PTR-TOF-MS season-average mixing ratios values presented in Figure 4 are also shown for reference (red circles)

The above Figure has been added to the supplement with relevant details (new Figure S7) and the following accompanying text added to the main manuscript at Line 364 (ACPD version of pdf):

"The results of the TD-GC-FID measurements in terms of the season-wise  hourly averaged values and corresponding minimum and maximum values observed for the same hourly interval alongwith the average PTR-TOF-MS values presented in Figure 4, are summarized in Figure S7. Even though the TD-GC-FID measurements present only a snapshot as the ambient sampling duration is shorter, the season-wise diel profiles are consistent with those obtained using the PTR-TOF-MS and the  average

mixing ratios obtained using the PTR-TOF-MS dataset also fall well within the range of mixing ratios observed using the TD-GC-FID. This provides further confidence in the high night-time isoprene observed during the post-monsoon season. The isoprene emissions at night during the post-monsoon season are likely due to combustion sources. Paddy residue burning and dung burning have the highest isoprene emission factors of ~0.2 g/kg ( Andreae 2019) and more than 8 Gg of isoprene is  released in the space of a few weeks during the post-monsoon season regionally from open paddy residue burning alone (Kumar et al., 2021). Previous studies from the region have also documented isoprene emissions from non-biogenic sources, which are active also at night (Kumar et al., 2020, Hakkim et al., 2021). In 2018 at another site in Delhi, using gas chromatography measurements made in pre-monsoon and post-monsoon, Bryant et al. 2023 reported average nocturnal mixing ratios of isoprene that were 5 times higher in the post-monsoon compared to the pre-monsoon and showed different diel profiles between the seasons. They found that the high night-time isoprene correlated well with carbon monoxide, a combustion tracer and suspected that in addition to the stagnant meteorological conditions, biomass burning sources could be a reason for the significant night time isoprene in Delhi in post-monsoon season and our findings using more comprehensive and high temporal resolution data further substantiate the surprising night-time isoprene."

We also checked out the interferences mentioned in the valuable study by Coggon et al. 2024.  As noted by Coggon et. al 2024 and in previous PTR-MS  reviews, the interference at m/z 69.067 (at which isoprene is detected) due to fragment ions from other compounds depends on the instrument operating conditions (Townsend ratio), instrument design and the mixture of VOCs present in ambient air while co-sampling isoprene. Coggon et al. 2024 very nicely clarified both these aspects and found that when influenced by cooking emissions and oil and natural gas emissions and at higher Townsend ratios, these interferences can be quite significant and even account for upto 50% of the measured signal attributed to isoprene in extreme cases. As noted in Table S1 of the orginal submission, we operated the PTR-TOF-MS at 120 Td which minimizes fragmentation even when it occurs, compared to when operated at 135-140 Td. Secondly the type of restaurant cooking emissions present in Las vegas and emissions over Oil and Natural Gas petrochemical facilities in USA for which Coggon et al. 2024 reported the highest isoprene interferences, are absent/neglible in the atmospheric environment of Delhi.  In the latter, open biomass burning sources such as paddy residue burning in post-monsoon season and garbage biomass fires that occur throughout the year are expected to play key roles. Even more specific GC based techniques have revealed that biomass combustion sources emit significant amounts of isoprene and furan. Our own supporting TD-GC-FID measurements alongwith those reported by Bryant et al. 2023 for post-monsoon season in Delhi, show that the night time isoprene in post-monsoon season is real. Concerning the interference on detection of acetaldehyde due to ethanol mentioned by the reviewer, we note that even in Coggon et al. 2024 this was reported to only be of significance in highly concentrated ethanol plumes such as those encountered on the Las-Vegas strip where ~1500 ppb of ethanol was present. On the contrary, in Delhi as noted in Table S2 of our orginal submission, ethanol values detected at m/z 47.076 were on average only 0.2 ppb (Interquartile range 0.16 ppb) during monsoon and 0.55 ppb (Interquartile range 0.5 ppb)  in post-monsoon season, respectively whereas acetaldehyde detected at m/z 45.03 was 3.34 and 7.75 ppb during monsoon and post-monsoon season, respectively.

This gives us confidence that  due to the different chemical environments, our attribution of isoprene, toluene and benzene in our PTR-TOF-MS data is accurate for Delhi.

Recognizing the importance of the  potential interferences mentioned in Coggon et al., 2024 , which are important to clarify so as to provide confidence in the measurements reported in our work from Delhi, we now also add the following text which includes discussion of the potential interferences mentioned in Coggon et al., 2024 after line 250 (ACPD version of pdf):

[revised manuscript text omitted]

Specific comments:

1. Title: I would recommend to change the title to "Observations of…" instead of "discovery of", since these compounds were not newly discovered but just observed in that location for the first time.

We appreciate the reviewer's comment. We had used the word discovery because although the compounds were reported previously, they have never been reported previously to occur collectively at the same site and that too under both clean (monsoon season) and extremely polluted conditions (post-monsoon season). The sources for each compound, namely methanethiol, dichlorobenzenes, C6-amides and C9-organic acids in the gas phase were also distinct pointing to industrial sources for the sulphur and chlorine containing compounds, photochemical source of the C6-amides and multiphase oxidation and chemical partitioning for the C9-organic acids. To our knowledge this is the first reported study world-wide to detect and observe only photo-chemically formed C6-amides in the gas phase.

Still heeding the reviewer1's and reviewer2's suggestion, we have now changed the title of the manuscript to :

"Reactive chlorine, sulphur and nitrogen containing volatile organic compounds impact atmospheric chemistry in the megacity of Delhi during both clean and extremely polluted seasons"

2. L 14 should be "COVID" capitalized

Thank you for pointing this out. In the revised version the necessary change has been made.

3. L 22 why was this surprising? These compounds have been observed in other urban areas, so I do not find this surprising.

We have removed the word "surprisingly", although to our knowledge, they have never been reported previously to occur collectively at the same site and that too under both clean (monsoon season) and extremely polluted conditions (post-monsoon season). The very limited prior reports are for individual compounds from different cities and mostly from source samples. It is also surprising that prior PTR-TOF-MS studies from Delhi (e.g. Jain et al. 2022; Wang et al., 2020) missed reporting these sulphur and chlorine compounds despite their unique isotope patterns that aid the identification.

4. L 33 should somewhere mention that Delhi is in India

Thank you for the suggestion. We have added India to the relevant sentence.

5. L 72-73: FAME and NCR are unexplained abbreviations

We appreciate the reviewer for pointing this out. These abbreviations have now been written in expanded form in the revised manuscript.

Faster Adoption and Manufacturing of hybrid and Electric vehicles (FAME)

National Capital Region (NCR)

6. Fig 1: a) the legend is illegible and b) the legend is too small

We appreciate the reviewer's feedback. We revised Figure 1 and the legend font size has been changed to 18 to 14 in Fig 1.

Revised Figure 1 is shown below for easy perusal and been added to the revised submission.

[Figure]

7.  Method section: Some more information of the inlet would be helpful. What was its length and diameter, was it heated for the whole length, what was the inlet residence time and flow rate?

Thank you for the helpful suggestion. The length of the inlet line was 5m and it was made of Teflon ( 3 m 1/8inch O.D. and 2 m 1/4inch O.D). The total inlet residence time was ~2.7 seconds. The part of the inlet that was indoors (3 m of 1/8 inch O.D.) was heated to 80 degree Celsius throughout the study period. We think this short inlet residence time and heated inlet facilitated the detection of IVOCs, relative to previous studies. As mentioned in the manuscript, the outer inlet was protected by a Teflon membrane particle filter (0.2 µm pore size, 47 mm diameter) to ensure that dust and debris did not enter the sampling inlet.

We have added the above additional information as new text after Line 170 (ACPD version of pdf):

"The length of the inlet line was 5m and made of Teflon (3m 1/8 inch O.D. and 2m 1/4inch O.D). The total inlet residence time was ~2.7 seconds. The part of the inlet that was indoors (3 m of 1/8 inch O.D.) was well insulated and heated to 80 degree Celsius. We think this short inlet residence time and heated inlet facilitated the detection of IVOCs, relative to previous studies."

L 117: the verbal description of color symbols is not necessary, these should be legible in the legend. Why are hospitals shown? Are they VOC sources?

Thank you for the suggestion. We have removed the mention of colors from the main text now. The font size of the legends has also been increased from 14 to 18 to make it more legible.

The hospitals in the maps are retained based on previously published literature (Bessonneau et al., 2013; Riveron et al., 2023), which suggest that medical facilities can act as a source of VOCs (e.g., acetone).

References:

Bessonneau, V., Mosqueron, L., Berrubé, A., Mukensturm, G., Buffet-Bataillon, S., Gangneux, J.P., Thomas, O.:VOC contamination in hospital, from stationary sampling of a large panel of compounds, in view of healthcare workers and patients exposure assessment. PLoS One 8 (2), e55535, https://doi.org/10.1371/journal.pone.0055535, 2013.

Riveron, T.P., Wilde, M.J., Ibrahim, W., Carr, L., Monks, P.S., Greening, N.J., Gaillard, E. A., Brightling, C.E., Siddiqui, S., Hansell, A.L., Cordell, R.L.:Characterisation of volatile organic compounds in hospital indoor air and exposure health risk determination. Build. Environ. 242, 110513 https://doi.org/10.1016/j.buildenv.2023, 2023

8. L 179 this sentence is unnecessary and sounds again like an advert.

In deference to the reviewer's suggestion, we have removed L179 from the revised version.

9. L 190 exchange "checked" with "compared"

Thank you for your suggestion. We have replaced "checked" with "compared" in the relevant text.

10. L 199: What about fragments?

We considered known fragmentation effects as well. The known fragments that were reported earlier at 120 Td, were considered and were mentioned in Table S2.

In the revised version we make this more clear as also stated in the response to a related point made earlier by the reviewer by adding the following new text after L199 (ACPD version pdf):

"The term "unambiguous" is used in the context of the accurate elemental composition/molecular formula assignment of the ions by leveraging the high mass resolution (8000-13000 over entire dynamic mass range) and detection sensitivity (reaching even 1 ppt or better for many ions; see Table S2) of the instrument. This enabled ensuring peaks due to expected isotopic signals were not construed as new compounds if their height was exactly as expected for a shoulder isotopic peak based on the natural abundance of isotopes of carbon, hydrogen, nitrogen, sulphur, chlorine and oxygen that made up the more abundant molecular ion. Where an ion could occur significantly due to fragmentation of another compound, the same has also been noted in Table S2 during attribution of the compound's name."

11. **L 208: Reproducible within how many %?**

Thank you for the comment. We note that the drifts in sensitivity were accounted for by the regular calibrations but it was further re-assuring to observe that calibration results were reproducible and better than 15% for all compounds other than trimenthyl benzene (18.7%) and

dichlorobenzene (20.6%). So overall, the reproducibility across the 8 calibration experiments carried out during the measurement campaign was ~20%.

Reproducibility percentages were calculated using the standard deviation of the average of the eight calibration experiments and are shown for easy perusal in the Table below.

| Protonated m/z | Compound | Reproducibility % |
|---|---|---|
| 33.03 | Methanol ($CH_4O$) | 13.5 |
| 42.03 | Acetonitrile ($C_2H_3N_1$) | 2.7 |
| 59.046 | Acetone ($C_3H_6O$) | 7.3 |
| 69.067 | Isoprene ($C_5H_8$) | 10.2 |
| 79.052 | Benzene ($C_6H_6$) | 5.6 |
| 93.069 | Toluene ($C_7H_8$) | 9.7 |
| 107.085 | Xylenes ($C_8H_{10}$) | 13.2 |
| 121.101 | 1,2,4-Trimethylbenzene ($C_9H_{12}$) | 18.7 |
| 146.977 | Dichlorobenzene ($C_6H_4Cl_2$) | 20.6 |

In the revised version we have added the following detail to Line 208 (ACPD pdf version) as follows:

"Results were reproducible (~21 % or better for all compounds) across all experiments ….."

12. **L 215: what is the overall uncertainty of the measured VOC concentrations? I assume it differs between compounds with a gas standard and compounds without one?**

Indeed. For compounds that were in the gas standard, the overall uncertainty calculated using the root mean square propagation of errors due to the accuracy of gas standard and flow controllers was ~13 % or better. For the rest of the compounds, it is estimated that the combined accuracy of the transmission function and the parameterized k-rates, put the overall uncertainty in the range of ±30% (Tobias et al., 2024). We have added this important information that was missing from the original submission and are grateful to the reviewer for drawing attention to it.

In the revised version we have added the following details after Line 215 (ACPD pdf version) as follows:

"The overall uncertainty calculated using the root mean square propagation of errors due to the accuracy of gas standard and flow controllers was ~13 % or better for compounds present in the VOC gas standard. For other compounds reported in this work, it is estimated that the combined accuracy of the transmission function and the parameterized k-rates, put the overall uncertainty in the range of ±30% (Reinecke et al., 2024)."


Thank you for the helpful suggestion. We have now revised Fig 6 and use the more similar slope to make the argument about the influence of additional sources of these compounds during the post-monsoon season. We have also added new Figures to the supplement which present the wind rose for these compounds for the same nighttime data from which the emission ratios were derived, which show that stronger influence of plumes during the post-monsoon season relative to the monsoon season can atleast partially explain the reason for the statistically different emission ratio observed during the post-monsoon season in Fig 6.
Revised Figure 6 is now shown below:

[Figure]

New wind rose plot added to the supplement is shown below:

[Figure]

Figure S8 Wind rose of night-time benzene, toluene, C8 and C9 aromatic compounds at the receptor site during monsoon (left column) and post monsoon (right column) season

The discussion after L 582 (per original ACPD version) has also been completely rewritten and strengthened as follows:

"It can be noted from Fig. 6, that during the monsoon season (from 18:00 to 23:00 and 00:00 to 06:00 local time) and post-monsoon season (18:00 to 23:00), the observed emission ratios as inferred from the slopes and fits are not statistically different from each other (all highlighted by oval circles) with values for benzene/CO, toluene/CO, sum of C8-aromatics , sum of C9-aromatics/CO in the range of 1.2-2.43, 3.14-6.76, 1.97-3.84, and 1.05-2.07, respectively. All these emission ratios fall within the range of what has been reported for typical petrol 2 and 4 wheeler vehicles in India in tail pipe emissions (Hakkim et al., 2021). For the monsoon season, although two linear fits are observed from 18:00 to 23:00 and 00:00 to 06:00, the values of the emission ratios as inferred from the respective slopes for all compounds overlap or are very close to each other and within the uncertainties for all compounds. We hypothesize that the two fits are due to the change in relative numbers of 2 wheelers and 4 wheelers. In the post-monsoon season however, for the time period in the second half of the night (00:00-06:00), the emission ratios derived from the slopes are statistically different from the ones observed in monsoon season and the first half of night in post-monsoon season (18:00-23:00). When we examined the wind rose plots for the same night-time data of the aforementioned compounds for each season (Figure S8), we noted that during the post-monsoon season more pollution plumes from the south east sector which has industrial facilities and the north west sector (a major fetch region for biomass burning plumes from regional paddy residue burning in Punjab and Haryana) occurred. During the post-monsoon season due to dip in temperatures at night, the heating demand (Awasthi et al., 2024) and associated open biomass burning (Hakkim et al., 2019) also goes up, relative to the monsoon period nights. Hence overall we think that these additional sources in the post-monsoon season, do add to the burden of these mainly traffic emitted aromatic compounds and could help explain atleast partially the higher emission ratios observed during the post-monsoon season ( 00:00- 06:00), wherein values for benzene/CO, toluene/CO, sum of C8-aromatics , sum of C9-aromatics/CO values range from 3.15-3.27, 7.72-8.68, 5.03-5.37, 2.6-2.76, respectively, and are statistically different from the others (ones marked by oval circles)."

26. Table 1: **Are the reported values really emission ratios or enhancement ratios? The emission ratio would depend on the distance from the source and the photochemical processing that has happened in between emission and observation. How to calculate emission ratios from ambient observations is described in (Gouw et al., 2017). Real observed emission ratios from flux observations in urban areas are shown in the SI of (Karl et al., 2018)**

The ones shown within parantheses in Table 1 are emission ratios , not enhancement ratios.

We note that we had already perused the valuable work of de Gouw et al., 2017 and even cited the same in the same section. We took care to calculate the emission ratios using only night-time data when chemical loss of these compounds is negligible as their main oxidation occurs through OH radicals during daytime, as also noted by de Gouw et al., 2017. Further, the other studies referred to in Table 1 for comparison also reported emission ratios derived using nighttime data.

Although the caption of Table 1 in original submission mentioned the same as under :

"Table 1: Comparative summary of the average mixing ratio (ppb) and Emission Ratios of VOC/ CO (ppb/ppm) of Delhi (in parentheses) with other megacities of Asia, Europe and North America."

Based on the reveiwer's comment, we see merit in clarifying this further in the text in the revised version.

Hence the following new text has been added after L 515 (original ACPD version) to make the above more clear:

"It may be further noted that we took care to calculate the emission ratios using only night-time data when chemical loss of these compounds is negligible as their main oxidation is through OH radicals during daytime, as also noted by de Gouw et al., 2017. Further, the other studies referred to in Table 1 for comparison, have also reported emission ratios derived using only nighttime data."

biomass burning in which more than 1 billion ton of biomass is burnt regionally (Kumar et al., 2021) within few weeks during mid October to end of November occur. This adds significantly to the atmospheric burden of these compounds, compared to megacities in developed countries where open biomass burning is better and more strictly regulated. Secondly, the meteorological conditions during post-monsoon season due to shallower boundary layer height and poor ventilation, and lack of wet scavenging due to absence of rain also slow down atmospheric removal of these compounds compared to megacities in Europe, wherein it rains more frequently throughout the year compared to Delhi."

Data availability: The data policy of ACP clearly states that data needs to be publicly available on a repository with a DOI. Availability upon request is no longer enough. (https://www.atmospheric-chemistry-and-physics.net/policies/data_policy.html)

Thank you for the suggestion. We agree and the data presented in manuscript will be publicly available as a Mendeley dataset as per the new data policy of ACP.

The data availability statement has been updated in the revised version.

---

## Author Comment (AC2)

**Discovery of reactive chlorine, sulphur and nitrogen containing ambient volatile organic compounds in the megacity of Delhi during both clean and extremely polluted seasons**

**Sachin Mishra et al. 2024**

We express our gratitude to Reviewer 2 for her/his careful and detailed reading of the manuscript and the insightful and helpful comments, which we were pleased to peruse. The helpful comments have contributed to improving the context, messaging and clarity of the original submission. Below, please find point-wise replies to the reviewer's comments and suggestions.

The point-wise replies (in blue) to the reviewer's comments (**in black**) are below. Changes made to the text are also provided after the replies (in red).

Reviewer #2:

Sachin Mishra et al.'s work detailed the measurement of volatile organic compounds (VOCs) in the urban environment of Delhi, India using the Proton Transfer Reaction Time of Flight Mass Spectrometer. The measurement period covered an extended period with "clean" and "polluted" monsoons that are impacted by several emission sources such as biomass burning events. The authors also argued the discovery of a few VOCs that were previously unaccounted in India. According to the authors, the major significance of the work was the speciation of new VOCs that might influence the urban atmospheric environments, particularly in the global south where fewer studies are reported compared to its counterpart. The application of new instrumentation to uncover underlying atmospheric chemical mechanisms is commendable.

We thank the reviewer for appreciating the results pertaining to speciation of new VOCs that influence urban atmospheric environments like Delhi and his/her encouraging remarks deeming the work commendable.

General Comments:

The research structure provided by the authors in its current version appears to be a measurement/instrument paper and/or regional research manuscript, in which the authors heavily detailed the capacity of IONICON's PTR-ToF-MS 10K. The paper reads as an atmospheric measurement technique study that highlights an instrumentation with improved resolution and sensitivity, although not developed by the same authors. I also believe that the authors failed to maximize the enhanced instrument capabilities based on the evidence provided in the manuscript, thus old versions of the PTR-ToF-MS can still capture the variability of VOCs discussed by the authors. A considerable portion of the manuscript was allotted to the discussion of the PTR-ToF-MS but with limited contribution of new information and implications that will improve the understanding of the state and behavior of the atmosphere and climate . The authors are highly recommended to present the study as a scientific paper with new and proper atmospheric

implication/s and dissociate it as methods and regional paper. More importantly, the authors should limit providing several grand claims with little to no supporting evidence. Significant manuscript revision and restructuring based on the following comments/suggestions is advised before publishing to ACP.

We appreciate the general feedback and comments. The motivation behind putting sufficient technical details was precisely because of the new instrumentation that has been applied for the study. We thought that the technical details would help readers have confidence in the results especially as we compare findings for routinely measured PTR-MS VOCs on which some pre-existing data exists albeit for shorter durations and pre COVID times using the older PTR-MS instrument versions and also extend the detection and quantification to new VOC compounds, leveraging the better detection limit and mass accuracy of PTR-ToF-MS 10K. Since Delhi is a complex megacity emission environment with high VOC complexity due to influence of biogenic, urban and episodic biomass fire emissions, it is necessary that the quality control and quality assurance are sufficiently well documented. We regret if it became a technique heavy study and below follow up point-wise on the helpful comments/suggestions of the reviewer to improve the discussion and presentation of the new scientific results in the revised version.

The most promising section of the manuscript is the discovery of methanethiol, dichlorobenzene, C6-amides, and C9 organic acids, which the authors attributed to the new instrumentation that was previously deployed in India. This is the main differentiation compared to prior VOC studies performed in India, where seasonal (monsoon vs post-monsoon) comparisons were already presented(Jain et al., 2022; Wang et al., 2020). The paper would have been more impactful if the authors focused on this section, instead of reporting previously detected VOCs (e.g. methanol, isoprene, etc.) in Delhi.

We thank the reviewer profusely for highlighting the study's novelty on account of discovery of methanethiol, dichlorobenzene, C6-amides, and C9 organic acids in ambient air, each possessing a different main source and occurring under both clean and polluted conditions at the same site.

We note that we referred to the valuable studies (Wang et al., 2020 and Jain et al. 2022) in the Introduction of the original submission (L65-L73 of original ACPD submission). However, Wang et al. 2020 did not present data from monsoon and post-monsoon seasons at all, their study shed light on the winter period (Dec-March) for the year 2018. Jain et al. (2022) did measure for some periods during January–February, May–July, and October–December of 2019, providing new insights but their measurement period did not include August and September at all, which are the main monsoon season months in Delhi. Further Jain et al. 2022 used a PTR-TOF-MS 8000 and had some limitations in terms of quality control (extensive assignments of $^{13}C$ isotopic peaks of hydrocarbons to reduced nitrogen compounds) which can be perused from the work.

1) Further in Jain et al., 2022, the background measurements to correct for instrument background were performed using a dry zero-air cylinder only every 2 weeks. Given how dirty the ambient air is, such a frequency may not always account for changes in instrumental background. In our work we collected the instrument background every 30 min for 5 min as stated in the original submission (L 171-173). No information was provided by Jain et al. (2022) concerning the overall uncertainty and detection limits of the measured compounds that could not be calibrated. They used a generic proton transfer reaction reaction rate of $2 \times 10^9$ $cm^3 s^{-1}$ for compounds that were not in the calibration standard. The mass resolution was at best ~5000 and detection limits even for the calibrated compounds were in the range of 40 ppt to 250 ppt, based on information about the same instrument (Sahu and Saxena, 2015; Sahu et al., 2017), as reported by the authors. The PTR-TOF-MS 10000 deployed in our work had significantly improved detection limits (reaching even 1 ppt or better for some compounds; see Table S2) and significantly better mass resolution (8000-13000) which enabled to detect and resolve the isotopic peaks for many compounds, also providing additional parameter for indentification of the exact elemental composition of the detected ion as illustrated using the example of furan ($C_4H_4O$) and triazole ($C_2H_3N_3$), earlier in reply to a comment by reviewer 1.

Thus, our work does provide a significant advancement in terms of quality of dataset even for the routinely measured PTR-MS VOCs over earlier works that used older PTR-MS versions and hence we have reported these in detail alongwith the newly discovered compounds in this atmospheric environment. Moreover, since all previous studies were carried out in pre-COVID period and significant policy measures were implemented (see L 71-74 of original ACPD version), this new study is timely and reflects current state of the atmospheric environment in Delhi-NCR for VOCs more realistically than the previous reports, which were also valuable for their insights when they were published.

References:

Sahu, L.K., Saxena, P., 2015. High time and mass resolved PTR-TOF-MS measurements of VOCs at an urban site of India during winter: role of anthropogenic, biomass burning, biogenic and photochemical sources. Atmos. Res. 164–165, 84–94. https://doi.org/10.1016/j.atmosres.2015.04.021.
Sahu, L.K., Tripathi, N., Yadav, R., 2017. Contribution of biogenic and photochemical sources to ambient VOCs during winter to summer transition at a semi-arid urban site in India. Environ. Pollut. 229, 595–606. https://doi.org/10.1016/j.envpol.2017.06.091.

At the current state of the manuscript, there are a lot of concerns that need to be addressed in the discussion of the four previously unaccounted VOCs. First, no statistical merits of these compounds were presented. What are the instrument blanks of these compounds? What are their corresponding limit of detection (L.O.D.) values? Are the values of such VOCs presented in the manuscript (i.e. ~50 ppt methanethiol in line 395) beyond the L.O.D? Instrumental background was measured according to the authors, but the results were never discussed in the main text and supplemental section.

We thank the reviewer for pointing out the missing details and agree that this would be helpful for readers and will strengthen the reported findings for these compounds. We deeply regret that some of this was missing from the original submission.

As kindly noted by the reviewer, the instrumental background was indeed measured. Below the Figure shows the measured raw 2-min temporal resolution data over a 3 h period for these four compounds, covering both the ambient and background measurement cycles. The first point after switching from ambient to background vale using the valve is affected by the transition is always excluded and hence not shown in the figure. It can be clearly seen that ambient levels of all the compounds are clearly above the instrument background, and that the instrument background is stable even in polluted post-monsoon season over several hours. We note that the average and and inter-quartile range for all the compounds were provided season-wise in Table S2 of the original submission. The mixing ratio of these four compounds was higher than the LoD values in both monsoon and post-monsoon seasons.

[Figure]

**Figure S3**: Example of measured data showing the instrumental backgrounds and ambient levels for methanethiol, dichlorobenzene, C-6 amide and C-9 carboxylic acid acid on 13.10.2022

In the revised version we have now added an additional column to Table S2 that lists the limit of detection (LoD) as well.

We have also added the following to the revised manuscript in Section 2.2 (after L200 of original submission).

"Table S2 also provides the limit of detection (LoD) of the compounds as well as the average and interquartile range observed season-wise for each ion. The LoD was calculated by taking the 2σ of the VOC-free zero air instrument background (Müller et al., 2014)."

References:

Müller, M., Mikoviny, T., Feil, S., Haidacher, S., Hanel, G., Hartungen, E., Jordan, A., Märk, L., Mutschlechner, P., Schottkowsky, R., Sulzer, P., Crawford, J. H., and Wisthaler, A.: A compact PTR-ToF-MS instrument for airborne measurements of volatile organic compounds at high spatiotemporal resolution, Atmos. Meas. Tech., 7, 3763–3772, https://doi.org/10.5194/amt-7-3763-2014 , 2014.

Besides the molecular attribution provided by IDA that is based on the closeness of the experimental and theoretical m/z, what are the other credible evidence of the authors in "unambiguously" identifying the four compounds in section 2.4?

Besides checking the closeness of the experimental and theoretical m/z values by IDA, we looked at the isotopic peaks for all 111 molecular formulae assigned in Table S1, including these four compounds. Due

to the LoD being better than 1 ppt for most ions, and the high mass resolution (8000-15000 across the dynamic mass range investigated), we could identify the isoptopic peaks and compare their observed abundance ( in percentage) with the theoretically calculated isotopic abundance percentage as a result of naturally occurring isotopic abundance of carbon, hydrogen, oxygen, nitrogen, chlorine and sulphur atoms. We found that the experimental isotopic percentages were in agreement with the theoretically calculated isotopic percentages for the 111 ions. This helped to rule out errors for the elemental composition assignment/molecular formula of ions that could also be candidates within the mass accuracy range, because the isotopic peak percentage would not match in case the ion had been assigned the wrong molecular formula. This also helped ensure that the isotopic peaks of more abundant compounds, e.g. C-13 isotopic peak of protonated furan ($C_4H_4O$; m/z = 73.0374) was not attributed to a highly reduced nitrogen containing compound like triazole ($C_2H_3N_3$).

The relevant mass spectra alongwith a Table for the isotopic peak match of these four compounds (methanethiol, C6-amide, dichlorobenzene, and C9 organic acid) is shown below for easy perusal.

[Figure]

**Figure S2** Mass spectra of methanethiol, dichlorobenzene, C6-amide and C9- carboxylic acid which also illustrate the high mass resolving power of the PTR-ToF-MS 10K enabling identification of isotopic peaks

| Molecular Formula | Protonated ion (m/z) $H^+$ | Observed Isotopic peaks | Theoretical Isotopic peaks |
|---|---|---|---|
| $CH_4S$ | 49.007 (M) (100%) | M+2 ($^{34}S$) (4.5%) | M+2 ($^{34}S$) (4.4%) |

| | | | |
|---|---|---|---|
| $C_6H_{13}NO$ | 116.108 (M) (100%) | M+1 ($^{13}C$) (6.5%) | M+1 ($^{13}C$) (6.5%) |
| $C_6H_4Cl_2$ | 146.977 (M) (100%) | M+2 ($^{37}Cl$) (64.9%) | M+2 ($^{37}Cl$) (64.8%) |
| $C_9H_{18}O_2$ | 159.140 (M) (100%) | M+1 ($^{13}C$) (9.9%) | M+1 ($^{13}C$) (9.7%) |

[Figure]

We have modified L200 of the orginal submission to incorporate this new information as follows:

"Fig S1 provides an example of visualization of mass spectra and peak assignment using the IDA software which also illustrate the high mass resolving power of the PTR-ToF-MS 10K, that enables separation of ion signals that differ by less than 0.04 Th, as well the identification of isotopic peaks of parent compounds like methanethiol, dichlorobenzene, C-6 amide and C-9 carboxylic acid acid (Fig S2), which are discussed in detail in Section 2.4."

and added the following new text:

"Example of measured data showing the instrumental backgrounds and ambient levels for methanethiol, dichlorobenzene, C-6 amide and C-9 carboxylic acid acid, over a 3h period are illustrated in Fig S3."

Did the authors use standard compounds to account the for the fragmentation pattern of these proposed compounds? Indeed, the PTR-ToF-MS is a soft-ionization technique, however, it also suffers from collision-induced dissociation (CID) that reduces the relative abundance/concentration of the primary molecular ion [M+H]. The relative contribution of acylium ions [M-OH] is more prominent in longer carboxylic acids, as observed in C2-C6 short-chain fatty acids (Hartungen et al., 2004). Thus, the C9 compound that the authors might not correspond to the real analyte measured in India.

We appreciate the reviewer's very valid concern and are grateful for the suggestion which has helped add an extra level of identification for attribution of m/z 159.14 to the C9-organic acid. Indeed as pointed out by Hartungen et al. (2004) and more recently by the insightful study of Salvador et al. (2022), carboxylic acids (RCOOH) can undergo dissociation reactions within the drift tube in addition to protonation, and form acylium ions as per the following reaction below (Hartungen et al., 2004):

$$H_3O^+ + RCOOH \rightarrow RCOOHH^+ \text{ (protonated ion)} + H_2O$$

$$H_3O^+ + RCOOH \rightarrow RCO^+ \text{ (acylium ion)} + 2\ H_2O$$

We also agree with the reviewer that the relative contribution of acylium ions would be significant for larger carboxylic acids and in response to this very valid concern did carry out an experiment using a carboxylic acid to ascertain the relative contribution under the same PTR-TOF-MS operating conditions (120 Td) used to acquire ambient data. While we could not do an experiment using nonanoic acid, we carried out the experiment using a dilute aqueous solution of octanoic acid ( a C8- carboxylic acid, which we also observed and reported in the original submission; see Table S2). The headspace of this was analysed in the instrument under the same operating conditions to ascertain the relative contribution of acylium and protonated ions and the laboratory experimental results were compared with that obtained in our ambient data for both the ions. The results are shown below:

Table below shows the relative abundance of C8-organic acid protonated ions and C8-acylium ions obtained during laboratory experiment and in the ambient data:

| Relative abundance | C8-organic acid protonated ion (m/z 145.123) | C8-acylium ion (m/z 127.112) |
|---|---|---|
| Experimental | 37% | 100% |
| Ambient | 46% | 100% |

Since the relative distribution obtained in the laboratory experiment and ambient data are reasonably close and within few percent of each, we updated the C8-carboxylic acid mixing ratios by considering sum of both ions for the ambient data and used the same approach to update the mixing ratios of C9-carboxylic acid.

In the measured ambient spectra, the corresponding acylium ion of C9-carboxylic acid ($C_8H_{17}CO^+$ detected at m/z 141.13) was present and correlated well with the protonated ion (r=0.83), which we had used for quantification of the C9-carboxylic acid previously. Example of the time series of both the measured ions in ambient air during monsoon and post-monsoon season are shown below for illustration:

[Figure]

Figure S9: Time series of C9-acylium fragment ion and C9-organic acid protonated ion during monsoon (25[th] Aug 2022 to 30[th] Aug 2022) and post monsoon season (27[th] Oct 2022 to 31[st] Oct 2022) provided for illustration

Hence the presence of the fragment ion has helped provide additional confirmation concerning the attribution of m/z 159.14 to the C9-organic acid and we are very grateful to the reviewer for the helpful suggestion. In the revised version we have updated the data for C9- carboxylic acid and C8-carboxylic acid (which was also listed in Table S2) to be more accurate, as earlier due to loss of  some of the signal to acylium ions, the values were underestimated.

Now we sum the signals due to the protonated and acylium ions and have updated the values in the text, Figure 5 and Table S2 with the new values and have added the following new discussion to Section 2.4 as follows:

"Hartungen et al. (2004) and more recently the insightful study by Salvador et al. (2022), have highlighted that carboxylic acids (RCOOH) can undergo dissociation reactions within the drift tube in addition to protonation, and form acylium ions as per the following reaction below (Hartungen et al., 2004) :

$H_3O^+ + RCOOH \rightarrow RCOOHH^+$ (protonated ion) $+ H_2O$

$H_3O^+ + RCOOH \rightarrow RCO^+$ (acylium ion) $+ 2\ H_2O$

We detected the corresponding acylium ion of C9-carboxylic acid ($C_8H_{17}CO^+$ detected at m/z 141.13) in the measured ambient spectra (Figure S9) and found that not only was it present but that it also correlated well in the ambient data with the protonated ion (r=0.83). The presence of the fragment ion and its correlation, provides additional confirmation concerning the attribution of m/z 159.14 to the C9-organic acid and for quantification the ion signals due to the protonated and acylium ions, were summed together for greater accuracy."

The authors also argued that the four compounds were only measured because of the extended volatility range mass spectrometer design and high sensitivity due to the ion booster and hexapole guide of the PTR-TOF-MS 10K system that enabled the detection of the compounds that are all intermediate volatility range organic compound (IVOC) (Line 387-388). What is the basis of the IVOC designation of these compounds? Did the authors calculate the saturation concentration of these compounds? Statements in Lines 387-388 also imply that the proper detection of these VOCs requires EVR, ion booster, and hexapole, whereas prior studies with older PTR-ToF-MS systems already detected IVOCs. For instance, Salvador et al., accounted IVOCs from the thermal desorption of organic aerosols (see Figure 2a of Salvador et al) using a PTR-ToF-MS 8000 without these new components (Salvador et al., 2022). Also, did the 10,000 m/Δm mass resolution, 1000 cps/ppbv sensitivity, and 1-10 pptv detection limit of the PTR-ToF-MS 10 K really help in the detection of these compounds? Do these compounds have known nominal mass interference and typical low concentration, thus requiring a highly mass-resolved and sensitive mass spectrometer? The authors should illustrate the mass spectra of these compounds, similar to Figure S1, to showcase the high resolving power of the instrument and to support their claims. The authors should heavily consider restating these statements.

Thank you for the comment. As mentioned in response to a previous point raised by the reviewer, we have already added illustrations of the mass spectra and ambient and instrument background examples in the form of new Figures(S1, S2 and S3) and added relevant text to the manuscript.

Below we further show how close some neighbouring ion peaks were for C6-amide, dichlorobenzene and C9-carboxylic acid, which also highlights that these compounds would all have suffered nominal mass intereference, that could not have been resolved using a lower mass resolution instrument. Further, given the low ambient abundance of few tens of ppt and the LoD of older PTR-MS instruments being in similar range and and lack of extended volatility sampling and analysis design, causing further losses within the inlet and mass spectrometer system, except for dichlorobenzene, it is doubtful if these compounds could have been detected in ambient air of Delhi, especially during the clean monsoon season. Indeed, none of the previous PTR-TOF MS studies from Delhi reported these compounds even though the sulfur and chlorine containing ones should have been possible to identify based on their unique isotope signatures.

We have replaced the old Figure S1 to show the relevance of the instrument's mass resolution to these particular compounds

[Figure]

**Figure S1** Example of mass spectra and peak assignment using IDA software which also illustrate the high mass resolving power of the PTR-ToF-MS 10K enabling separation of ion signals that land at the same nominal masses.

As regards the basis for the IVOC classification, we did calculate the saturation mass concentration ($C_0$) of methanethiol, C6-amide, dichlorobenzene, and C9 organic acids using the method described in Li et al. 2016 using the following equation:

$$C_0 = \frac{M\ 10^6\ p_0}{760\ R\ T} \tag{1}$$

wherein M is the molar mass [g mol$^{-1}$], R is the ideal gas constant [8.205 x 10$^{-5}$ atm K$^{-1}$ mol$^{-1}$ m$^3$], $p_0$ is the saturation vapor pressure [mm Hg], and T is the temperature (K). Organic compounds with $C_0 > 3$ x $10^6$ µg m$^{-3}$ are classified as VOCs while compounds with $300 < C_0 < 3$ x $10^6$ µg m$^{-3}$ as Intermediate VOCs (IVOCs).

The table below shows the saturation mass concentration ($C_0$) and the class of these compounds based on the saturation mass concentration ($C_0$) values.

| Protonated m/z | Compounds | Molecular formula | $C_0$ | Class |
|---|---|---|---|---|
| 49.007 | Methanethiol | $CH_4S$ | $4.0 \times 10^9$ | VOC |
| 116.108 | C6 amide | $C_6H_{13}NO$ | $7.8 \times 10^4$ | IVOC |

| 146.977 | Dichlorobenzene | $C_6H_4Cl_2$ | $7.9 \times 10^2$ | IVOC |
|---------|-----------------|--------------|-------------------|------|
| 159.14 | C9 organic acid | $C_9H_{18}O_2$ | $2.3 \times 10^5$ | IVOC |

We have now added the above missing pertinent information to the revised version at L 385 as follows:

"The saturation mass concentrations ($C_0$) of methanethiol, C6-amide, dichlorobenzene, and C9 organic acids were calculated using the method described in Li et al. 2016 using the following equation:

$$C_0 = \frac{M\ 10^6\ p_0}{760\ R\ T}$$
(1)

wherein M is the molar mass [g mol$^{-1}$], R is the ideal gas constant [8.205 x 10$^{-5}$ atm K$^{-1}$ mol$^{-1}$ m$^3$], $p_0$ is the saturation vapor pressure [mm Hg], and T is the temperature (K). Organic compounds with $C_0 > 3$ x $10^6$ µg m$^{-3}$ are classified as VOCs while compounds with $300 < C_0 < 3$ x $10^6$ µg m$^{-3}$ as Intermediate VOCs (IVOCs)."

The recent pioneering TD-PTR-TOF-MS study by Salvador et al. (2022) measured the IVOCs by thermally desorbing the aerosol particles at different temperatures after collection on filter paper for more than 8 h. The concentrations of the IVOCs after this preconcentration were generally high enough (100 ngm$^{-3}$ - 300 ngm$^{-3}$) in that study to be detected by PTR-TOF-MS 8000, as even 50 ng m$^{-3}$ of the larger compounds would typically be above the detection limits of 10 ppt and not lower. However, in our study, the IVOCs were detected in the gas phase directly in real time without thermal desorption of pre-collection/pre-concentration on filters for 8 h. The latter therefore can only provide averaged ambient levels at temporal resolution of 8 h, so we do think that detection of gas phase IVOCs in ambient air without preconcentration and thermal desorption may not have been possible for most of the IVOCs and VOCs due to their low abundance, and was feasible in our study due to the extended volatility range design and high sensitivity of the instrument provided by the ion booster and hexapole guide.

References:

Li, Y., Pöschl, U., and Shiraiwa, M.: Molecular corridors and parameterizations of volatility in the chemical evolution of organic aerosols, Atmos. Chem. Phys., 16, 3327–3344, https://doi.org/10.5194/acp-16-3327-2016 , 2016.

Salvador, C. M., Chou, C. C. K., Ho, T. T., Ku, I. T., Tsai, C. Y., Tsao, T. M., Tsai, M. J., and Su, T. C.: Extensive urban air pollution footprint evidenced by submicron organic aerosols molecular composition, npj Climate and Atmospheric Science, 5, 96, 10.1038/s41612-022-00314-x, 2022.

The authors relied heavily on the distinction between ''clean" monsoon and "polluted" post-monsoon season to explain their observation, even for VOCs that shouldn't respond evidently to such atmospheric event changes. For instance, the emission of biogenic compounds such as isoprene are known to be strongly dependent on temperature, UV, and vegetation activities. The authors should properly account for the sources of the VOCs. In line 361, the authors even argued that the increased biomass burning and traffic controlled the abundance of isoprene. Any supporting study and/or study to support such a claim?

We have already addressed this point in detail in reply to a similar comment ny reviewer 1. For completeness sake and convenience of reviewer 2 , we are reproducing the same below:

We appreciate the reviewer's suggestions, especially since we also reported night time high isoprene during the post-monsoon season, which may be surprising when compared to many ambient studies conducted in Europe and the USA, where isoprene is emitted primarily in the daytime from biogenic sources as a function of temperature and radiation. Besides considering the potential interferences due to fragmentation of higher molecular weight compounds as had already been flagged in previous

PTR-MS reviews, we took additional measures to confirm the identity and diel profiles for the major compounds mentioned by the reviewer such as isoprene, benzene and toluene. Thus during our campaign whole air samples from the rooftop of the same building were collected into 6 L passivated SilcoCan air sampling steel canisters (Restek) (Kumar et al., 2020; Vettikkat et al., 2020, Shabin et al., 2024) to measure isoprene, toluene and benzene using a thermal desorption gas chromatograph coupled to a flame ionisation detection. Samples were collected during post monsoon season, four times a day [during start hourly intervals corresponding to 08:00, 13:00, 18:00, 20:00 local time; n=39] and during monsoon season also four times a day during the following hourly intervals [during start hourly intervals corresponding to: 00:00, 07:00, 12:00, 19:00 local time; n=15]. The results of the TD-GC-FID measurements in terms of season-wise hourly averaged values and corresponding minimum and maximum values observed for the same hourly interval are shown in the graph below alongwith PTR-TOF-MS data as reported in our original submission.

[Figure]

**Figure S7:** Season-wise hourly averaged average mixing ratios of isoprene, benzene and toluene measured using the TD-GC-FID are shown as diamonds and minimum and maximum values for that hourly sampling interval as shaded regions ( in green for isoprene, pink for benzene and purple for toluene). The PTR-TOF-MS season-average mixing ratios values presented in Figure 4 (red circles) During our campaign whole air samples from the rooftop of the same building were collected into 6 L

passivated SilcoCan air sampling steel canisters (Restek) (Kumar et al., 2020; Vettikkat et al., 2020, Shabin et al., 2024) to measure isoprene, toluene and benzene using a thermal desorption gas chromatograph coupled to a flame ionisation detector (TD-GC-FID). Technical details pertaining to the TD-GC-FID measurements are available in Shabin et al., 2024. Air samples were collected near the PTR-TOF-MS inlet on the rooftop during post monsoon season, four times a day [during hourly intervals corresponding to 08:00, 13:00, 18:00, 20:00 local time (n=39,) and during monsoon season also four times a day during hourly intervals corresponding to: 00:00, 07:00, 12:00, 19:00 local time (n=15).

The above Figure has been added to the supplement as Figure S7 and the following accompanying text added to the main manuscript at Line 364 (ACPD version of pdf):

"The results of the TD-GC-FID measurements along with the average PTR-TOF-MS values presented in Figure 4, are summarized in Figure S7. Even though the TD-GC-FID measurements present only a snapshot as the ambient sampling duration is shorter, the season-wise diel profiles are consistent with those obtained using the PTR-TOF-MS and the average mixing ratios obtained using the PTR-TOF-MS dataset also fall well within the range of mixing ratios observed using the TD-GC-FID. This provides further confidence in the high night-time isoprene observed during the post-monsoon season. The isoprene emissions at night during the post-monsoon season are likely due to combustion sources. Paddy residue burning and dung burning have the highest isoprene emission factors of ~0.2 g/kg (Andreae 2019) and more than 8 Gg of isoprene is released in the space of a few weeks during the post-monsoon season regionally from open paddy residue burning alone (Kumar et al., 2021). Previous studies from the region have also documented isoprene emissions from non-biogenic sources, which are active also at night (Kumar et al., 2020, Hakkim et al., 2021). In 2018 at another site in Delhi, using gas chromatography measurements made in pre-monsoon and post-monsoon, Bryant et al. 2023 reported average nocturnal mixing ratios of isoprene that were 5 times higher in the post-monsoon compared to the pre-monsoon and showed different diel profiles between the seasons. They found that the high night-time isoprene correlated well with carbon monoxide, a combustion tracer and suspected that in addition to the stagnant meteorological conditions, biomass burning sources could be a reason for the significant night time isoprene in Delhi in post-monsoon season and our findings using more comprehensive and high temporal resolution data further substantiate the surprising night-time isoprene."

Specific and technical comments:

1. The title is misleading, particularly the word "discovery". Analysis or a similar term might be more appropriate

We appreciate the reviewer's comment. We had used the word discovery because although the compounds were reported previously, they have never been reported previously to occur collectively at the same site and that too under both clean (monsoon season) and extremely polluted conditions (post-monsoon season). The sources for each compound, namely methanethiol, dichlorobenzenes, C6-amides and C9-organic acids in the gas phase were also distinct pointing to industrial sources for the sulphur and chlorine containing compounds, photochemical source of the C6-amides and multiphase oxidation and chemical partitioning for the C9-organic acids. To our knowledge this is the first reported study world-wide to detect and observe only photo-chemically formed C6-amides in the gas phase.

Still heeding the reviewer1's and reviewer2's suggestion, we have now changed the title of the manuscript to :

*"Reactive chlorine, sulphur and nitrogen containing volatile organic compounds impact atmospheric chemistry in the megacity of Delhi during both clean and extremely polluted seasons"*

2. Line 18: the term "unambiguously" is inappropriate for this case, particularly with the lack of standard compounds and/or supporting measurements (e.g. GC-MS) that will confirm the identities of the proposed compounds. I understand the PTR-ToF-MS 10K has an enhanced mass resolution, but molecular formula alone cannot be the sole evidence for the "unambiguously" identification of VOCs.

We regret the confusion in the original submission due to the context of the use of the word "unambiguous" not be explicitly stated. As noted in detailed reply to reviewer 1, we have already clarified the same and now in the revised version clarified it as follows:

Appreciating that the meaning of this was not clear enough in the orginal submission and also in response to the point mentioned concerning fragmentation at L199 later, we have now added the following text to make it more clear in the revised version after L199-200 ( ACPD version of pdf):

"The term "unambiguous" is used in the context of the accurate elemental composition/molecular formula assignment of the ions by leveraging the high mass resolution (8000-13000 over entire dynamic mass range) and detection sensitivity (reaching even 1 ppt or better for many ions; see Table S2) of the instrument. This enabled ensuring peaks due to expected isotopic signals were not construed as new compounds if their height was exactly as expected for a shoulder isotopic peak based on the natural abundance of isotopes of carbon, hydrogen, nitrogen, sulphur, chlorine and oxygen that made up the more abundant molecular ion. Where an ion could occur significantly due to fragmentation of another compound, the same has also been noted in Table S2 during attribution of the compound's name."

Line 70: **What are NCR and FAME? An important note: the authors argue that these VOC data sets are unique since they were collected post-COVID, in which new programs were introduced that will impact air quality. However, nowhere in the succeeding section were these accounted for or even mentioned, except for an overarching statement in the conclusion that has no strong foundation.**

NCR stands for National Capital Region (NCR) and FAME stands for Faster Adoption and Manufacturing of hybrid and Electric Vehicles (FAME).

To address the reviewer's suggestion to provide more information concerning policy changes implemented after COVID lockdowns of 2020, we have made the following additions:

L70: The associated text has been re-written with additional details as follows:

"We note that all these were pre-COVID period datasets, and that since these observations many new regulations have been put in place e.g. for traffic with the introduction of BS-VI (EURO6 equivalent) in 2020 and the Faster Adoption and Manufacturing of hybrid and Electric vehicles (FAME) program for promotion of E-vehicles, and for industries with a ban on the use of petcoke in the National Capital Region (NCR) and the crackdown on unregistered industries (Guttikunda et al, 2023). After COVID lock-downs happened in 2020, a new Commission for Air Quality Management in Delhi National Capital Region and its Adjoining Areas (CAQM) was set up in November 2020 (https://caqm.nic.in/index.aspx?langid=1 ). Under its mandate, depending on air quality level, it promulgates immediate graded response action plans (GRAP; https://caqm.nic.in/index1.aspx?lsid=4168&lev=2&lid=4171&langid=1) that instruct civic authorities to shut-down or restrict particular emission sources. Furthermore on 7 August 2020, the Delhi government announced a new Delhi Electric Vehicle (EV) Policy. In order to address the high-upfront cost of EVs (ICE vehicles), the Delhi EV Policy provides demand incentives for purchasing electric vehicles. The incentives help bring cost parity for EVs and are in addition to those outlined in the Faster Adoption and Manufacturing of Hybrid and Electric Vehicles (FAME II). In the budget

allocation for 2020, the Government of India allocated $600 million USD for clean air measures through the Ministry of Housing and Urban Affairs (MoHUA) to 46 cities across India. These have been detailed in the report by Arpan Chatterji (2020). Thus overall, important changes to the transport emission sector, construction and urban industrial sector and residential sector were implemented at a policy level after 2020 to reduce air pollution in the Delhi-NCR region."

Reference: Arpan Chatterji, "Air Pollution in Delhi: Filling the Policy Gaps,"

ORF Occasional Paper No. 291, December 2020, Observer Research Foundation.

3. Line 122: Atmospheric ventilation is not a common meteorological parameter, and thus requires further explanation. What is the physical meaning/basis of atmospheric ventilation?

Thank you for your suggestion.

We have added the following explanation to make this more clear in the revised version:

"Atmospheric ventilation or ventilation coefficient (VC) is a good proxy for the dilution and dispersion of air pollutants near the surface (Hakkim et al., 2019). It is defined as the product of boundary layer height (m) and wind speed (ms$^{-1}$). The VC represents the rate at which air within the mixed layer is transported away from a region of interest and provides information about how concentrations of pollutants are modulated through transport of air over that region."

Reference:

Hakkim, H., Sinha, V., Chandra, B. P., Kumar, A., Mishra, A. K., Sinha, B., Sharma, G., Pawar, H., Sohpaul, B., Ghude, S.D., Pithani, P., Kulkarni, R., Jenamani, R.K., and Rajeevan, M.: Volatile organic compound measurements point to fog-induced biomass burning feedback to air quality in the megacity of Delhi, Sci. Total Environ., 689, 295-304, https://doi.org/10.1016/j.scitotenv.2019.06.438 , 2019.

4. Line 137: Section 2.2 is too long with unnecessary information regarding the technique. This is not the first study that utilized this technique/instrument. Authors should consider moving them to supplement file or totally removing them.

We have removed some text but in response to the both reviewers' comments asking for additional technical clarifications could not shorten the overall length.

5. Line 206: The authors indicated that the instrument was calibrated eight times during the extended measurement period, which should account for the drift/changes/responses in the instrumentation. However, the results of the calibration were not presented, even at least in the supplement file. This information is important, as it would provide confidence in the dataset that the variability in the VOC concentration can be accounted primarily for changes in atmospheric events(monsoon vs post-monsoon), instead of the instrument signal drift.

Thank you for the comment, which was also made by reviewer 1.We note that the drifts in sensitivity were accounted for by the regular calibrations but it was further re-assuring to observe that calibration results were reproducible and better than 15% for all compounds other than trimenthyl benzene (18.7%)

and dichlorobenzene (20.6%). So overall, the reproducibility acorss the 8 calibration experiments spread of the measurement campaign was ~20%.

Reproducibility percentages were calculated using the standard deviation of the average of the eight calibration experiments and are shown for easy perusal in the Table below.

| Protonated m/z | Compound | Reproducibility % |
|---|---|---|
| 33.03 | Methanol ($CH_4O$) | 13.5 |
| 42.03 | Acetonitrile ($C_2H_3N_1$) | 2.7 |
| 59.046 | Acetone ($C_3H_6O$) | 7.3 |
| 69.067 | Isoprene ($C_5H_8$) | 10.2 |
| 79.052 | Benzene ($C_6H_6$) | 5.6 |
| 93.069 | Toluene ($C_7H_8$) | 9.7 |
| 107.085 | Xylenes ($C_8H_{10}$) | 13.2 |
| 121.101 | 1,2,4-Trimethylbenzene ($C_9H_{12}$) | 18.7 |
| 146.977 | Dichlorobenzene ($C_6H_4Cl_2$) | 20.6 |

In the revised version we have added the following detail to Line 208 (ACPD pdf version) as follows:

"Results were reproducible (~21 % or better for all compounds) across all experiments ….."

Figure S5 of revised supplement also shows some example calibration plots.

6. Lines 229-255: These statements should be moved to the experimental section. Please note that several sentences were already mentioned and should not be repeated. The authors should consider providing a sample ion that was identified through the proposed method of the authors to visually guide the readers.

We appreciate the reviewer's suggestion and have shifted the text to a new section "2.3 Mass assignment and compound identification" in revised manuscript. We have also added a visual representation of the method for a sample ion (m/z 146.997; dichlorobenzene) and have added the Figure below to the revised supplement as Figure S6.

[Figure]

**Figure S6** Visual representation of the process providing also an example compound attribution to an ion (for m/z 146.997; dichlorobenzene) showing isotopic peak match with theoretically predicted isotopic abundance

7.   Figure 3: The baseline concentrations of some VOCs were evidently different for both monsoon and post-monsoon seasons. An obvious example is the isoprene, particularly during mid-October when baseline concentration is increasing even at nighttime while a clear flat baseline is observed during clean monsoon. The same insights can be applied to acetonitrile and acetaldehyde. Any possible explanation? Does this impact the measured concentration between the two periods?

8.   Figure 4: The nighttime enhancement of isoprene during post-monsoon is questionable. As a biogenic VOC with a short atmospheric lifetime, isoprene typically only has daytime enhancement. Is this related to the increasing baseline indicated in comment 8?

As rightly noted by the reviewer, the baseline concentration of VOCs like isoprene, acetaldehyde, and acetonitrile are high in post monsoon as compared to monsoon season. We wish to re-iterate that the time series of the VOCs shown in Fig 3 is background-corrected, which means the zero air background instrumental response for these VOCs has already been subtracted from the ambient data. During the post monsoon season, large scale agricultural waste burning and enhanced biomass burning occurred and this can also be observed from the increase in the daily fire count data presented in Fig 3 (top panel). The concerns pertaining to the night time isoprene have been addressed in detail while replying to the earlier general comments of reviewer 1 and earlier in this response. The same are mentioned again for convenience of the reviewer.

We also checked out the interferences mentioned in the valuable study by Coggon et al. 2024.  As noted by Coggon et. al 2024 and in previous PTR-MS  reviews, the interference at m/z 69.067 (at which isoprene is detected) due to fragment ions from other compounds depends on the instrument operating conditions (Townsend ratio), instrument design and the mixture of VOCs present in ambient air while co-sampling isoprene. Coggon et al. 2024 very nicely clarified both these aspects and found that when influenced by cooking emissions and oil and natural gas emissions and at higher Townsend ratios, these interferences can be quite significant and even account for upto 50% of the measured signal attributed to isoprene in extreme cases. As noted in Table S1 of the orginal submission, we operated the PTR-TOF-MS at 120 Td which minimizes fragmentation even when it occurs, compared to when operated at 135-140 Td. Secondly the type of restaurant cooking emissions present in Las vegas and emissions over Oil and Natural Gas petrochemical facilities in USA for which Coggon et al. 2024 reported the highest isoprene interferences, are absent/neglible in the atmospheric environment of Delhi.  In the latter, open biomass burning sources such as paddy residue burning in post-monsoon season and garbage biomass fires that occur throughout the year are expected to play key roles. Even more specific GC based techniques have revealed that biomass combustion sources emit significant amounts of

isoprene and furan. Our own supporting TD-GC-FID measurements alongwith those reported by Bryant et al. 2023 for post-monsoon season in Delhi, show that the night time isoprene in post-monsoon season is real. Concerning the interference on detection of acetaldehyde due to ethanol mentioned by the reviewer, we note that even in Coggon et al. 2024 this was reported to only be of significance in highly concentrated ethanol plumes such as those encountered on the Las-Vegas strip where ~1500 ppb of ethanol was present. On the contrary, in Delhi as noted in Table S2 of our orginal submission, ethanol values detected at m/z 47.076 were on average only 0.2 ppb (Interquartile range 0.16 ppb) during monsoon and 0.55 ppb (Interquartile range 0.5 ppb) in post-monsoon season, respectively whereas acetaldehyde detected at m/z 45.03 was 3.34 and 7.75 ppb during monsoon and post-monsoon season, respectively.

This gives us confidence that due to the different chemical environments, our attribution of isoprene, toluene and benzene in our PTR-TOF-MS data is accurate for Delhi.

For convenience we reproduce below new text has been added to this section as also stated in the earlier reply to the relevant section:

"The results of the TD-GC-FID measurements along with the average PTR-TOF-MS values presented in Figure 4, are summarized in Figure S7. Even though the TD-GC-FID measurements present only a snapshot as the ambient sampling duration is shorter, the season-wise diel profiles are consistent with those obtained using the PTR-TOF-MS and the average mixing ratios obtained using the PTR-TOF-MS dataset also fall well within the range of mixing ratios observed using the TD-GC-FID. This provides further confidence in the high night-time isoprene observed during the post-monsoon season. The isoprene emissions at night during the post-monsoon season are likely due to combustion sources. Paddy residue burning and dung burning have the highest isoprene emission factors of ~0.2 g/kg (Andreae 2019) and more than 8 Gg of isoprene is released in the space of a few weeks during the post-monsoon season regionally from open paddy residue burning alone (Kumar et al., 2021). Previous studies from the region have also documented isoprene emissions from non-biogenic sources, which are active also at night (Kumar et al., 2020, Hakkim et al., 2021). In 2018 at another site in Delhi, using gas chromatography measurements made in pre-monsoon and post-monsoon, Bryant et al. 2023 reported average nocturnal mixing ratios of isoprene that were 5 times higher in the post-monsoon compared to the pre-monsoon and showed different diel profiles between the seasons. They found that the high night-time isoprene correlated well with carbon monoxide, a combustion tracer and suspected that in addition to the stagnant meteorological conditions, biomass burning sources could be a reason for the significant night time isoprene in Delhi in post-monsoon season and our findings using more comprehensive and high temporal resolution data further substantiate the surprising night-time isoprene."

9. Line 339: What does CNG mean?

   CNG means Compressed Natural Gas.

   The full form has been added to the revised version.

10. Figure 5: The C6 amide is missing a superscript. The same goes for line 447.

    We could not find that any missing superscript in line 447 or Figure 5.

11. Lines 411-417: The possible sources of methanethiol were based on assumptions even though the authors have enough data to possibly explain their results. Is there a particular wind direction where

methanethiol is more dominant? Does this coincide with the possible locations of the manufacturing facilities?

We thank the reviewer for this helpful suggestion. This will indeed strengthen the source provenance. As shown in Figure 5 of the original submission, the elevated methanethiol values in the windrose had a clear directional dependence from the north to south east sector. This is the region where various manufacturing facilities and industrial areas of Delhi like Patparganj (north-east) Okhla, Faridabad (south) are situated and have been also been marked in the Figure 1 b.

We have added the above relevant discussion to the revised version after L417 of orginal submission as follows:

"Figure 5 confirms that elevated methanethiol values in the windrose had a clear directional dependence from the area spanning the north to south east sector. This is the region where various manufacturing facilities and industrial areas of Delhi like Patparganj (north-east) Okhla, Faridabad (south) are situated and these industrial estates were earlier marked in Figure 1b."

Line 422-426: How did the authors arrive at such results? Do the authors have a relevant data set showing the enhancement of organic aerosols due to the methanethiol in Delhi? Similarly, do the authors have evidence of the interaction of organic haze with sulfur compounds such as methanethiol? Consider removing such statements.

We thank the reviewer for the comment which will improve the discussion. In response to the same, we have replaced the following text which may have been worded too strongly in the orginal submission and have added an additional reference:

"……provide evidence that sulphur and carbon chemistry coupling can impact the organic haze and atmospheric sulphur chemistry in planetary atmospheres, and to our knowledge the present study presents the first evidence from a polluted megacity supporting the hypothesis (Reed et al., 2020)."

The revised statement now reads as follows

"More recently, Reed et al. (2020) performed laboratory experiments and observed that even trace amounts of organosulphur compounds, such as thiols and sulfides, can significantly enhance the organic aerosol mass concentration and its particle effective density. Though there has not been any relevant data set attributing the enhancement of organic aerosols to methanethiol in Delhi specifically, previous studies have found enhanced secondary aerosol formation rates during haze and fog episodes (Acharja et al., 2022). These studies collectively suggest an increase in the haze events in Delhi is linked to sulphur chemistry in which methanethiol due to its high reactivity and atmospheric chemistry could also be a contributor along with ammonia and other sulphur containing molecules."

Reference:

Acharja, P., Ali, K., Ghude, S. D., Sinha, V., Sinha, B., Kulkarni, R., Gultepe, I., and Rajeevan, M. N.: Enhanced secondary aerosol formation driven by excess ammonia during fog episodes in Delhi, India, Chemosphere 289, 133155, https://doi.org/10.1016/j.chemosphere.2021.133155, 2022.

Reed, N. W., Browne, E. C., and Tolbert, M. A.: Impact of hydrogen sulfide on photochemical haze formation in methane/nitrogen atmospheres, ACS Earth and Space Chemistry, 4(6), 897-904, https://doi.org/10.1021/acsearthspacechem.0c00086 , 2020.

12. Lines 428-431: These statements are unclear. The increasing sales of methanethiol are not relevant to the few ppt concentrations observed in Delhi.

We apologize for this and have now removed the L428-431 from the revised manuscript.

13. Line 475-477: What is the physical meaning of "inverse of daytime relative humidity" and how does it indicate that it partitions back and forth between the gas phase and aerosol phase? How did the authors arrive at such results? Please note that temperature and relative humidity have an indirect relationship in most cases. The authors might be following the trend of temperature instead. Also, the daytime enhancement profile of the C9 acid is a clear indication that it has a photochemical source and I am not sure why the authors are arguing that biomass burning and "evaporation" from the aerosol phase is the major source of the C9 acid.

We appreciate the reviewer's concern. The values shown correspond to RH dropping from ~90% to 50%. At 90% humidity hygroscopic aerosol particles tend to have a wet diameter that is 1.5 to 2 times larger than their dry diameter. This reduces to ~1.05-1.2 times the dry diameter by the time RH drops to 50%. These changes in the aerosol aqueous phase are expected to impact the partitioning between gas and aerosol phase for water soluble compounds. The known temperature dependence of the Henry's Law constant of organic acids may further amplify this effect (Khan et al. 1995). The C9- acid profile does not really match well time-wise with that of a photochemical oxidation product in the data. For reference we show the diel profile of isoprene, its oxidation product (sum of methyl vinyl ketone and methacrolein; MVK+MACR) and the C9-carboxylic acid for the monsoon period below:

[Figure]

It can clearly be discerned from the above that the C9-carboxylic acid has a delayed increase in daytime peak relative to MVK+MACR, which rise after their precursor isoprene, and co-incide with the radiation peak.

Temperature and relative humidity (RH) do indeed anti-correlate with lower RH at higher temperature and vice-versa typically being the case. RH represents that ratio between amount of water vapour an air parcel has, compared to the maximum amount of water vapour it can hold at that temperature (saturation vapour pressure). However the saturation vapor pressure follows an exponential curve. Assuming equilibrium conditions between gas and aerosol phase, when RH is high, water soluble compounds like carboxylic acids can dissolve and partition to aerosol phase resulting in reduced gas phase concentrations and the partitioning is primarily controlled by the volume of the available aerosol aqueous phase (which is controlled by RH) and secondarily the temperature dependenc of Henry's law coefficient, if there is no major change in absolute humidity.

Below we show the plot for the C9-carboxylic acid versus ambient temperature:

[Figure]

It can be seen above that the relationship is exponential between ambient levels of C9- carboxylic acid and temperature but linear with inverse of RH as shown in Figure 5. Hence, the "inverse of daytime relative humidity" was used as a better parameter to indicate the partitioning of the C9-compound back and forth between the gas phase and aerosol phase.

Reference:

Khan, I., Brimblecombe, P., and Clegg, S. L., Solubilities of Pyruvic Acid and the Lower (C1-C6) Carboxylic Acids. Experimental Determination of Equilibrium Vapour Pressures Above Pure Aqueous and Salt Solutions. Journal of Atmospheric Chemistry 22: 285-302, 1995.

14. Line 500-506 and Figure 6: The statistical merits (slope and r2) for both monsoon and post-monsoon do not have evident different values yet the authors came up with unsupported claims and implications (i.e. additional sources).

Line 538-539: Again, how did the authors arrive at such conclusion (i.e., the influence of non-traffic sources) based on the comparison of emission factors?

Based on the helpful suggestion by reviewer1 and in response to above point by reviewer2, we have now revised Fig 6 and use the more similar slope to make the argument about the influence of additional sources of these compounds during the post-monsoon season. We have also added new Figures to the supplement which present the wind rose for these compounds for the same nighttime data from which the emission ratios were derived, which show that a stronger influence of plumes during the post-monsoon season relative to the monsoon season can atleast partially explain the reason for the statistically different additional slope, observed during the post-monsoon season in Fig 6.

Thus discussion after L 582 (per original ACPD version) has also been completely rewritten and strengthened as follows:

"It can be noted from Fig. 6, that during the monsoon season (from 18:00 to 23:00 and 00:00 to 06:00 local time) and post-monsoon season (18:00 to 23:00), the observed emission ratios as inferred from the slopes and fits are not statistically different from each other (all highlighted by oval circles) with values for benzene/CO, toluene/CO, sum of C8-aromatics, sum of C9-aromatics/CO in the range of 1.2-2.43, 3.14-6.76, 1.97-3.84, and 1.05-2.07, respectively. All these emission ratios fall within the range of what has been reported for typical petrol 2 and 4 wheeler vehicles in India in tail pipe emissions (Hakkim et al., 2021). For the monsoon season, although two linear fits are observed from 18:00 to 23:00 and 00:00 to 06:00, the values of the emission ratios as inferred from the respective slopes for all compounds overlap or are very close to each other and within the uncertainties for all compounds. We hypothesize that the two fits are due to the change in relative numbers of 2 wheelers and 4 wheelers. In the post-monsoon season however, for the time period in the second half of the night (00:00-06:00), the emission ratios derived from the slopes are statistically different from the ones observed in monsoon season and the first half of night in post-monsoon season (18:00-23:00). When we examined the wind rose plots for the same night-time data of the aforementioned compounds for each season (Figure S8), we noted that during the post-monsoon season more pollution plumes from the south east sector which has industrial facilities and the north west sector (a major fetch region for biomass burning plumes from regional paddy residue burning in Punjab and Haryana) occurred. During the post-monsoon season due to dip in temperatures at night, the heating demand (Awasthi et al., 2024) and associated open biomass burning (Hakkim et al., 2019) also goes up, relative to the monsoon period nights. Hence overall we think that these additional sources in the post-monsoon season, do add to the burden of these mainly traffic emitted aromatic compounds and could help explain atleast partially the higher emission ratios observed during the post-monsoon season ( 00:00- 06:00), wherein values for benzene/CO, toluene/CO, sum of C8-aromatics , sum of C9-aromatics/CO values range from 3.15-3.27, 7.72-8.68, 5.03-5.37, 2.6-2.76, respectively, and are statistically different from the others (ones marked by oval circles)."

Revised Figure 6 is now shown below:

[Figure]

**Figure 1: Emission ratios (VOC (ppb)/CO (ppm)) of benzene, toluene, C8 aromatics and C9 aromatics for both monsoon (left panel) and post monsoon (right panel) periods respectively. The data points for each period are colour coded with the hour of the day (18:00 L.T to 06:00 L.T).**

We have added new wind rose plot added to the supplement is shown below:

[Figure]

**Figure S8 Wind rose of night-time benzene, toluene, C8 and C9 aromatic compounds at the receptor site during monsoon (left column) and post monsoon (right column) season.**

15. Line 543: Unprecedented is inappropriate here given the prior VOC comparison studies performed in Delhi.

We note that we have used the word "unprecedented" in the specific context as conveyed by the relevant sentence which is:

"This study has provided *unprecedented characterization of the VOC chemical composition* of ambient air in Delhi *for the clean monsoon and extremely polluted post-monsoon seasons*."

We note that we referred to the previous valuable studies (Wang et al., 2020 and Jain et al. 2022) in the Introduction of the original submission (L65-L73 of original ACPD submission). However, Wang et al. 2020 did not present data from monsoon and post-monsoon seasons at all , their study shed light on the winter period (Dec-March) for the year 2018. Jain et al. (2022) did measure for some periods during January–February, May–July, and October–December of 2019, providing new insights but their measurement period did not include August and September at all, which are the main monsoon season months in Delhi. Further Jain et al. 2022 used a PTR-TOF-MS 8000 and had some limitations in terms of quality control (E.g. extensive assignment of the $^{13}C$ isotopic peaks to reduced nitrogen compounds) which can be perused from their work by the reviewer.

Further in Jain et al., 2022, the background measurements to correct for instrument background were performed using a dry zero-air cylinder only every 2 weeks. Given how dirty the ambient air is, such a frequency may not always account for changes in instrumental background. In our work we collected the instrument background every 30 min for 5 min as stated in the original submission (L 171-173). No information was provided by Jain et al. (2022) concerning the overall uncertainty and detection limits of the measured compounds that could not be calibrated. They used a generic proton transfer reaction reaction rate of 2 x 109 cm3 s-1 for compounds that were not in the calibration standard. The mass resolution was at best ~5000 and detection limits even for the calibrated compounds were in the range of 40 ppt to 250 ppt, based on information about the same instrument (Sahu and Saxena, 2015; Sahu et al., 2017), as reported by the authors. The PTR-TOF-MS 10000 deployed in our work had significantly improved detection limits (reaching even 1 ppt or better for some compounds; see Table S2) and significantly better mass resolution (8000-13000) which enabled to detect and resolve the isotopic peaks for many compounds, also providing additional parameter for indentification of the exact elemental composition of the detected ion as illustrated using the example of furan (C4H4O) and triazole (C2H3N3), earlier in reply to a comment by reviewer 1.

Therefore, we respectfully disagree with the reviewer because although previous VOC studies have been performed in Delhi including by our own group, none comes close to the comprehensive VOC characterization reported in this work for the monsoon and post-monsoon seasons (also in terms of data coverage). The present work also reported new compounds that have never been reported from South Asia previously and VOCs that have only rarely been observed in other atmospheric environments. Hence for aforementioned reasons, we feel that use of the word "unprecedented" is justified.

16. Line 551: Why provide again the molecular formula and name? It was already mentioned in the results and discussion.

We thank the reviewer for this helpful edit.

We have now removed the molecular formula from the relevant text in the revised version.

17. The authors keep on arguing that their compounds are IVOCs without presenting information regarding the volatility/saturation concentration of these compounds. I suggest that the authors present

or cite these values or calculate them using the identified molecular formula. See Mohr et al., 2019 for reference (Mohr et al., 2019)

As clairified in an earlier reply, we have added the above missing pertinent information for this to the revised version at L 385 as follows:

"The saturation mass concentration ($C_0$) of methanethiol, C6-amide, dichlorobenzene, and C9 organic acids were calculated using the method described in Li et al. 2016 using the following equation:

$$C_0 = \frac{M\ 10^6\ p_0}{760\ R\ T} \qquad (1)$$

wherein M is the molar mass [g mol$^{-1}$], R is the ideal gas constant [8.205 x 10$^{-5}$ atm K$^{-1}$ mol$^{-1}$ m$^3$], $p_0$ is the saturation vapor pressure [mm Hg], and T is the temperature (K). Organic compounds with $C_0 > 3$ x $10^6$ µg m$^{-3}$ are classified as VOCs while compounds with $300 < C_0 < 3$ x $10^6$ µg m$^{-3}$ as Intermediate VOCs (IVOCs)."

18. Line 567: Where's the PM2.5 data to support such a claim?
    We appreciate the reviewer's criticism.

    The relevant sentence L 567 was:
    "The reactive gaseous organics were found to rival the high mass concentrations of the main air pollutant in exceedance at this time, namely PM$_{2.5}$ during the extremely polluted periods."

    We now provide the time series of the PM$_{2.5}$ hourly data alongwith acetonitrile ( a biomass burning VOC tracer) as a new Figure in the revised supplement, which was measured at the same site using a BAM sensor (Shabin et al., 2024) and have modified L 567 as follows:
    "The reactive gaseous organics, which reached total averaged mass concentrations of ~85 µg m$^{-3}$ (monsoon season) and ~265 µg m$^{-3}$ (post-monsoon season) were found to rival the high mass concentrations of the main air pollutant in exceedance at this time, namely PM$_{2.5}$ during the extremely polluted periods (post-monsoon season average: ~145 µg m$^{-3}$ which exceeds the 24h national ambient air quality standard of 60 µg m$^{-3}$). The data of the time series of the PM$_{2.5}$ hourly data alongwith acetonitrile ( a biomass burning VOC tracer) measured at the same site is provided in Figure S10"

[Figure]

Figure S10: Time series of hourly PM$_{2.5}$ and acetonitrile measured at IMD Delhi site

19. Line 573-576: The authors should be careful in asserting such statements (i.e. not as state of the art). Other techniques such as GC-MS, Orbitrap, and CIMS which have improved mass resolution and sensitivity can provide better quantification of the VOCs compared to the instrument utilized by the authors. I commend the expanded list of VOCs measured by PTR-ToF-MS 10K, however, the compounds listed by the authors are already measured in previous studies using other techniques that lead to a better understanding of the complex atmospheric systems across the globe.

We appreciate the reviewer's point and have removed mention of the phrase "not as state of the art" in L573-576.

The modified sentence now reads as follows:

"All previous VOC studies in the literature from a dynamically growing and changing megacity like Delhi were reported for periods before 2020 (pre-COVID) times, without the new enhanced volatility VOC quantification technology deployed for the first time in a complex ambient environment of a developing world megacity like Delhi."

---

## Author Response (AR2)

**Reactive chlorine, sulphur and nitrogen containing volatile organic compounds impact atmospheric chemistry in the megacity of Delhi during both clean and extremely polluted seasons**

**Sachin Mishra et al. 2024**

We thank the editor and reviewers for their kind comments on the revised version. Below, please find point-wise replies to the minor remaining reviewers' comments. We hope these revisions are now adequate for the manuscript to be accepted in ACP.

The point-wise replies (in blue) to the reviewer's comments (**in black**) are below. Changes made to the text are also provided after the replies (in red).

**Response to Anonymous referee #1**

I thank the authors for their thorough review of the manuscript, and I think the current version is greatly improved. There are just two very minor points that I would like to see revised.

We thank the reviewer for deeming the revised submission to be greatly improved!

There are just two very minor points that I would like to see revised.

1) In their response to my comment no. 29 (L.552/ L. 759 in the new manuscript version with tracked changes), the authors revised their text as follows" "The detection of new compounds that have previously not been discovered in Delhi's air, under both the clean and polluted periods such as methanethiol, dichlorobenzenes, C6-amides and C9-organic acids in the gas phase was very surprising, considering there have been several PTR-TOF MS studies earlier (Wang et al., 2020; Tripathi et al., 2022; Jain et al., 2022). Our data points to both industrial sources of the sulphur and chlorine compounds, photochemical source of the C6-amides and multiphase oxidation and chemical partitioning for the C9-organic acids."

a) The authors should refrain from "discovered" and instead use the more neutral word "observed". (Similarly, the heading of section 3.4 should be revised. I would be happy with the wording "first observation in Delhi" but the connotation of "discovery" is for me the first time in the world.)

b) "there have been several PTR-TOF-MS studies earlier" should be clarified as meaning studies in Delhi.

c) I am missing a hypothesis as to why previous studies have not seen these compounds. Could it be that they were measured but not reported (after all, not everyone reports or identifies all peaks detected) - maybe due to improvements in analysis software? Or could it be that the location or year caused differences in the nearby emission sources? I assume the previous studies were done in different places in Delhi. Are any of the reported potential sources nearby the measurement location in this study?

Reply: Out of regard for the reviewer's minor comments in 1a and 1b above, we have revised the relevant text which now reads as follows:

"The detection of new compounds that have previously not been observed in Delhi's air, under both the clean and polluted periods such as methanethiol, dichlorobenzenes, C6-amides and C9-organic acids in the gas phase was very surprising, considering there have been several PTR-TOF MS studies in Delhi earlier (Wang et al., 2020; Tripathi et al., 2022; Jain et al., 2022). Our data points to both industrial sources of the sulphur and chlorine compounds, photochemical source of the C6-amides and multiphase oxidation and chemical partitioning for the C9-organic acids."

Concerning the hypothesis on why previous PTR-TOF-MS studies did not report these compounds, as highlighted in our detailed replies to reviewer 2. We are mentioning the main ones below for convenience.

We think the previous PTR-TOF-MS studies had significant limitations in terms of quality control of the dataset as well as some technical limitations of the instruments used (all were PTR-TOF-MS 8000).

For example in Jain et al., 2022, the background measurements to correct for instrument background were performed using a dry zero-air cylinder only every 2 weeks. Given how dirty the ambient air in Delhi can get, such a frequency may not always account for changes in instrumental background. In our work we collected the instrument background every 30 min for 5 min as stated in the original submission (L 171-173). Further no information was provided by Jain et al. (2022) concerning the overall uncertainty and detection limits of the measured compounds that could not be calibrated. They used a generic proton transfer reaction reaction rate of $2 \times 10^9$ cm$^3$ s$^{-1}$ for compounds that were not in the calibration standard. All the previous works used a PTR-TOF-MS 8000, the mass resolution of which was at best ~5000 and detection limits even for the calibrated compounds were in the range of 40 ppt to 250 ppt, based on information about the instrument as reported by the authors (Sahu and Saxena, 2015; Sahu et al., 2017). Finally their mass spectral assignment had issues e.g. extensive assignments of $^{13}$C isotopic peaks of hydrocarbons to reduced nitrogen compounds were made, which can be perused from the works and also as pointed out in our replies to the queries raised by reviewers during the interactive discussion.

In terms of technical limitations of the PTR-TOF-MS instruments deployed in those previous works the following pertinent to note:

The PTR-TOF-MS 10000 deployed in our work had significantly better detection limits (reaching even 0.3 ppt for some compounds) and significantly higher mass resolution (8000-13000) which enabled us to detect and resolve the isotopic peaks for many compounds, thus providing an additional parameter for identification and quantification of methanethiol, C6-amide, dichlorobenzene and C-9 organic acids. It is unlikely that they did not detect the compounds because of site differences, because one of them was actually deployed in the same campus and same building of IMD as our deployment. We can also rule out local emission source because we did not observe high values of methanethiol and dicholorobenzene only at low wind speeds but rather from a specific fetch region which is known to have multiple industries and is located east and south of the measurement site.

2) In L. 645 the revised paper mentions the "insightful study by Salvador et al.". I think that, although Salvador et al. would surely be delighted by this assessment, judgments like "insightful" should be refrained from in a scientific paper.

Thank you for the comment. We agree and have now removed the adjective "insightful" from the relevant sentence in the newly revised version.

**Response to Anonymous referee #2**

Comment: The authors still failed to respond properly to reviewer 2, specific comment#7 regarding the "increasing" baseline of some of the VOCs indicated in figure 3. The authors should consider applying a baseline correction procedure that will reduce the baseline into zero to low signal, thus providing accurate concentration of the VOCs.

Thank you for the comment. The relevant comment#7 of reviewer 2 is being reproduced below for convenience again:

Comment #7.    Figure 3: "The baseline concentrations of some VOCs were evidently different for both monsoon and post-monsoon seasons. An obvious example is the isoprene, particularly during mid-October when baseline concentration is increasing even at nighttime while a clear flat baseline is observed during clean monsoon. The same insights can be applied to acetonitrile and acetaldehyde. Any possible explanation? Does this impact the measured concentration between the two periods?"

We regret that our previous reply was not clear to the reviewer. Here we would like to reiterate that these are not chamber measurements, where while measuring the same matrix, instrumental backgrounds can result in shifting of baseline values but measuremments made in ambient air. Further:

1) We can rule out completely that a shifting "instrumental" baseline has any effect on the ambinet profile time series shown in Figure 3, which have been carefully corrected for any changes in the "instrumental background" through stringent quality control. As stated previously in the manuscript (original submission (L 171-173), in our work we collected the instrument background every 30 min for 5 min and the reported VOC mixing ratios have been "background" subtracted. Furthermore, in response to the reviewer's concern we also showed how low the instrument backgrounds typically were in Fig S3 of the revised supplement using the 2 min averaged data.

2) The reason for higher "baseline" values in the time series of Fig 3 during the polluted post-monsoon season is the increased ambient pollution due to enhanced biomass burning emissions and reduced ventilation coeffient in the polluted post-monsoon season. This leads to an increase in the regional nighttime background values of compounds co-emitted by biomass burning such as isoprene, acetaldehyde and acetonitrile. These have already been justified by comparison with complementary TD-GC-FID measurements of isoprene and toluene and benzene in addition to the PTR-TOF-MS in the revised manuscript.

   We hope the above conclusively clarifies the concern of reviewer 2.

References:

Jain, V., Tripathi, S.N., Tripathi, N., Sahu, L.K., Gaddamidi, S., Shukla, A.K., Bhattu, D., and Ganguly, D.: Seasonal variability and source apportionment of non-methane VOCs using PTR-TOF-MS measurements in Delhi, India, Atmos. Environ. 283, 119163, https://doi.org/10.1016/j.atmosenv.2022.119163, 2022.
Sahu, L.K., Saxena, P., 2015. High time and mass resolved PTR-TOF-MS measurements of VOCs at an urban site of India during winter: role of anthropogenic, biomass burning, biogenic and photochemical sources. Atmos. Res. 164–165, 84–94. https://doi.org/10.1016/j.atmosres.2015.04.021.
Sahu, L.K., Tripathi, N., Yadav, R., 2017. Contribution of biogenic and photochemical sources to ambient VOCs during winter to summer transition at a semi-arid urban site in India. Environ. Pollut. 229, 595–606. https://doi.org/10.1016/j.envpol.2017.06.091.